# Defining the relationship between maternal care behavior and sensory development in Wistar rats: Auditory periphery development, eye opening and brain gene expression

Jingyun Qiu[1]☯, Preethi Singh[1]☯, Geng Pan[1]☯, Annalisa de Paolis[2], Frances A. Champagne[3], Jia Liu[4], Luis Cardoso[2], Adrián Rodríguez-Contreras☯[1]*

1 Department of Biology and Center for Discovery and Innovation, City College, City University of New York, New York, New York, United States of America, 2 Department of Biomedical Engineering, City College, City University of New York, New York, New York, United States of America, 3 Department of Psychology, University of Texas at Austin, Austin, Texas, United States of America, 4 Neuroscience Initiative, Advanced Science Research Center at the Graduate Center, City University of New York, New York, New York, United States of America

☯ These authors contributed equally to this work.
* arodriguezcontreras@ccny.cuny.edu

**Data Availability Statement:** All relevant data are within the paper and its supporting information files.

## Abstract

Defining the relationship between maternal care, sensory development and brain gene expression in neonates is important to understand the impact of environmental challenges during sensitive periods in early life. In this study, we used a selection approach to test the hypothesis that variation in maternal licking and grooming (LG) during the first week of life influences sensory development in Wistar rat pups. We tracked the onset of the auditory brainstem response (ABR), the timing of eye opening (EO), middle ear development with micro-CT X-ray tomography, and used qRT-PCR to monitor changes in gene expression of the hypoxia-sensitive pathway and neurotrophin signaling in pups reared by low-LG or high-LG dams. The results show the first evidence that the transcription of genes involved in the hypoxia-sensitive pathway and neurotrophin signaling is regulated during separate sensitive periods that occur before and after hearing onset, respectively. Although the timing of ABR onset, EO, and the relative mRNA levels of genes involved in the hypoxia-sensitive pathway did not differ between pups from different LG groups, we found statistically significant increases in the relative mRNA levels of four genes involved in neurotrophin signaling in auditory brain regions from pups of different LG backgrounds. These results suggest that sensitivity to hypoxic challenge might be widespread in the auditory system of neonate rats before hearing onset, and that maternal LG may affect the transcription of genes involved in experience-dependent neuroplasticity.

**Funding:** AR-C Grant SC1DC015907 National Institutes of Health URL: https://www.nih.gov/ The funders had no role in study design, data collection and analysis, decision to publish, or preparation of the manuscript.

**Competing interests:** The authors have declared that no competing interests exist.

## Introduction

In several mammalian species, including humans, maternal care is the main source of nutritional, social, and sensory stimulation that is important for survival and has the potential to impact the neurobiological development of the offspring [1, 2]. Variation in rat postpartum maternal licking and grooming (LG) has been used as a model to select dams with individual differences in LG behavior and study the developmental re-programming of the offspring's adult stress response [3–12]. However, characterization of the effects of the rearing experiences provided by dams with different LG behavior is incomplete, particularly with respect to how maternal LG stimulation may affect sensory development of the offspring during sensitive periods of early postnatal development, when various environmental challenges can severely disrupt mother infant interactions, reduce the chances of survival, and cause severe long-term neurobiological deficits in the progeny [13, 14].

Oxygen shortage is the exemplar environmental factor responsible for neonatal encephalopathy in preterm and term neonates. In a rat model of experimental perinatal anoxia, the frequency of LG from surrogate dams towards manipulated pups was not different compared to a control condition [15]. However, female pups exposed to anoxia showed deficits in maternal LG and pup retrieval as adults [16]. Neonatal encephalopathy has been linked to hearing loss and communication deficits such as poor speech discrimination [17]. Experimental and clinical studies have also demonstrated a strong susceptibility of brainstem auditory neurons to asphyxia and hypoxic-ischemic insult, but the molecular mechanisms for this sensitivity are not well understood [18–22]. Oxygen homeostasis is regulated by hypoxia-inducible factor 1a (Hif1a), which is involved in regulating the transcription of target genes that mediate the adaptation of cells and tissues to oxygen shortage [23]. Intriguingly, homeostatic responses to oxygen shortage involve changes in energy supply, cell proliferation and cell survival, which overlap with functions assigned to neurotrophin signaling in the auditory system [24]. Immunohistochemical evidence in rodents suggests that neurotrophin signaling increases first in the cochlea, and subsequently, in brainstem and cortical auditory regions during a period that begins one week before hearing onset and ramps up during the first month of life [24–27]. In contrast, Hif1a mRNA expression has been identified in cochlear explants and dissociated cochlear preparations of one-week old rats, but it has not been investigated in the central auditory system during postnatal development [28]. The limited availability of gene expression screens from multiple auditory brain regions represents a major challenge to understand the relationship between different signaling pathways and auditory system development. Defining the relationship between maternal LG, neurodevelopment and gene expression in the auditory system could help develop treatments for auditory and communication deficits caused by oxygen shortage in neonates.

In the present study, we tested the hypothesis that variation in maternal LG during the first week of life is associated with neurodevelopmental changes in the auditory system of Wistar rat pups. We performed tests of auditory brainstem response (ABR), tracked eye opening (EO), imaged development of the middle ear cavity using micro-CT X-ray tomography, and monitored gene expression changes in auditory brainstem and primary sensory cortex of pups reared by low-LG or high-LG dams. The results provide the first evidence that transcription of genes in the hypoxia-sensitive pathway and neurotrophin signaling is regulated during separate sensitive periods that occur before and after hearing onset, respectively. Although the timing of ABR onset, EO, and the relative mRNA levels of genes involved in the hypoxia-sensitive pathway did not differ between pups from different LG groups, we found statistically significant differences in the relative mRNA levels of *Ngf*, *Akt1*, *Sort1 and Nfkb1* in auditory brain regions from pups of different LG backgrounds. These results suggest that sensitivity to

hypoxic challenge is widespread in the auditory system of neonate rats before hearing onset, and that maternal LG may affect the transcription of genes involved in experience-dependent neuroplasticity.

## Materials and methods

### Experimental design and statistical analyses

Fig 1 illustrates three groups of experiments designed to test the main hypothesis of this study: A) developmental tracking, B) combined functional/structural analysis, and C) gene expression screen in pups reared by low-LG or high-LG dams. Table 1 summarizes the number of adult animals used for breeding, the number of dams used in maternal behavior selection experiments, and the number of dams selected for each group of experiments. Table 2 summarizes the number and demographics of all pups used for developmental tracking and functional/structural experiments. Auditory brainstem and sensory cortex gene expression was examined in a total of 21 pups from either sex. We compared 4 developmental stages comprising birth (P0; n = 3 pups, each from different litters); the end of the first postnatal week at P7 (n = 3 low-LG pups from two litters, and 3 high-LG pups from one litter); the end of the second postnatal week at P15 (n = 3 low-LG pups from two litters, and 3 high-LG pups from one litter); and the weaning age at P21 (n = 3 low-LG pups, each from different litters, and 3 high-LG pups from two litters).

Unless indicated, data represent mean ± SD. Statistical analyses were done with Prism 6 software (GraphPad). When appropriate, data sets were tested for normality using the D'Agostino and Pearson omnibus K2 test. For data sets that passed the normality test, means were analyzed with an ordinary one-way ANOVA and the Holm-Sidak's multiple comparisons test. For data sets that did not pass the normality test, medians were analyzed with the ANOVA

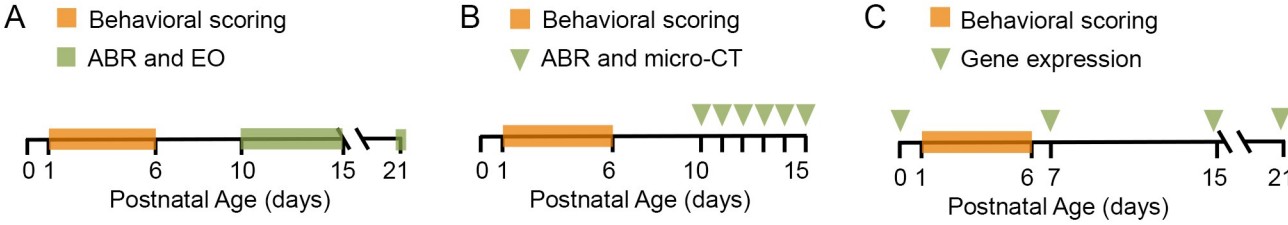

**Fig 1. Experimental approach to test the hypothesis that maternal licking and grooming (LG) influences sensory neurodevelopment in Wistar rat pups. A**, Developmental tracking experiments. After maternal behavior scoring and selection between P1 and P6, the auditory brainstem response (ABR) and eye opening (EO) were tracked daily in all pups from selected litters between P10 and P15, and at P21. **B**, Structure/function experiments. Following maternal behavioral scoring and selection between P1 and P6, correlative ABR and micro-CT X-ray (micro-CT) tomography measurements were obtained in pups from selected litters between P10 and P15 (indicated by arrow heads). **C**, Gene expression experiments. Pup brain samples were processed for mRNA extraction and gene expression screen at P0, P7, P15 and P21 (indicated by arrowheads).

**Table 1. Number of adult rats used in experiments.**

| Cohort | Breeding | | Dam | Developmental Tracking | | Functional/Structural | | Gene Expression | |
|---|---|---|---|---|---|---|---|---|---|
| | Males | Females | Selection | Low-LG | High-LG | Low-LG | High-LG | Low-LG | High-LG |
| 1 | 20 | 40 | 36 | 1 | 2 | 1 | 1 | 3 | 4 |
| 2 | 20 | 40 | 33 | 1 | 4 | - | - | - | - |
| 3 | 20 | 40 | 32 | 2 | 1 | - | - | - | - |
| 4 | 20 | 40 | 32 | 3 | 3 | 1 | 1 | 1 | 1 |
| Total | 80 | 160 | 133 | 7 | 10 | 2 | 2 | 4 | 5 |

**Table 2. Number of litters and pups used in developmental tracking and functional/structural experiments.**

| Cohort | Selected litters | | Number of pups | | Percent Males | | Percent Females | |
|---|---|---|---|---|---|---|---|---|
| | Low-LG | High-LG | Low-LG | High-LG | Low-LG | High-LG | Low-LG | High-LG |
| 1 | 2 | 3 | 26 | 46 | 73 | 50 | 27 | 50 |
| 2 | 1 | 4 | 13 | 45 | 31 | 49 | 69 | 51 |
| 3 | 2 | 1 | 24 | 11 | 54 | 27 | 46 | 73 |
| 4 | 4 | 4 | 45 | 45 | 51 | 44 | 49 | 56 |
| Total | 9 | 12 | 108 | 147 | 55 | 46 | 45 | 54 |

and the Kruskal-Wallis multiple comparisons test. P values were adjusted for multiple comparisons using software built in algorithms. Gene expression data was analyzed by the ordinary one-way ANOVA and Fisher's multiple comparisons test. Alpha = 0.05 was used to denote significance when testing for statistical differences between means or medians.

## Animal housing and breeding

The Institutional Animal Care and Use Committee of the City College of New York specifically reviewed and approved this study. The four cohorts of adult Wistar rats used in this study were obtained from a commercial supplier at postnatal age 65 (P65, Charles River). Rats were kept in a controlled environment at 22˚C with an alternating 12 h light and dark cycle (lights were on at 7:00 hrs and off at 19:00 hrs). Water and food were available *ad libitum*. Male and female Wistar rats at postnatal day 65 were obtained from a commercial supplier (Charles River). Upon arrival, same sex rat pairs were housed in Plexiglas cages and acclimated to the animal care facility for one week. After acclimation, simple randomization with shuffled cage numbers was used to assign single males to a cage with a female pair. Breeding trios were housed together for five days. At the completion of the breeding period males were removed from the study and female pairs were housed together for 14 more days. Wistar rats have a gestation period of 22 days. Hence, 19 days after mating females were housed individually in Plexiglas cages that were supplied with paper towels as nesting material. Cages were checked for births everyday at 9:00 hrs, 12:00 hrs, and 17:00 hrs. On the day of birth (P0), pups were weighted and dams and their litters were placed in clean Plexiglas cages. Females that were not pregnant were removed from the study. Cages were undisturbed during behavioral scoring between P1 and P6, and routine twice per week cage cleaning resumed after behavioral scoring was finished. At P8, dams and their litters were transported to a satellite room for acclimation before testing.

## Maternal behavior scoring and selection criteria

Methods for scoring maternal behavior and litter selection were adapted from a previous study [29]. In brief, five 1-hour maternal behavior observation sessions were performed by trained observers daily at 6:00 hrs, 9:00 hrs, 13:00 hrs, 19:00 hrs, and 21:00 hrs between pup ages P1 to P6. Every 1-hour observation consisted of 3 minute-long bins where the following behaviors were scored if observed: no contact with pups, contact with pups, dam is drinking, dam is eating, dam is self-grooming, dam is nest building, dam is licking and grooming (LG) pups in the anogenital or body region, and various levels of arched-back nursing described previously [29]. Unless indicated, LG scores in this study represent the frequency of LG in 100 observations per day (60 observations in the light cycle and 40 observations in the dark cycle) expressed as percent LG per day, or as average percent LG obtained from six-day scores. For

every cohort, LG histograms were generated and individual dams were selected if their six-day average LG score was 1 SD above (high-LG) or 1 SD below (low-LG) the six-day average LG score of their cohort. Dams and litters that were not selected were removed from the study.

### Developmental tracking

Gender, body weight, onset of the auditory brainstem response (ABR) and eye opening (EO) were tracked. Pup weight was recorded at P0, daily between P10 and P15, and at P21. Pup gender was determined between P10 and P15 using as joint criteria the anogenital distance and the presence or absence of multiple nipples to distinguish between males and females. EO was determined between P10 and P21, and was scored if at least one eyelid was open.

### ABR tracking

All ABR measurements were done blind to LG group. ABR measurements were obtained daily between P10 and P15, and at P21. Anesthesia was induced inside a Plexiglas chamber with 3–5% isoflurane and maintained through a nose cone with 1.5% isoflurane dissolved in medical grade oxygen (gas flow set at 1 L min$^{-1}$). ABR measurements were performed inside a double wall sound attenuated room (IAC). Anesthetized pups were placed onto a heating pad set at 37˚C to keep them warm throughout the procedure. Subdermal electrodes were placed behind the right ear (reference electrode), at the vertex (active electrode), and at the left shoulder (ground electrode). A calibrated electrostatic Kanetec MB-FX free field speaker was used to deliver click sounds at 40 Hz with intensities ranging from 102 to 2 dB sound pressure level (SPL) in 5 dB decrements. Clicks were synthesized with TDT system 3 hardware (Tucker-Davis Technologies), and presented at 20 kHz with alternating polarity to minimize the presence of stimulus artifacts. Speaker calibration was done with a type 7012 ½ inch ACO Pacific microphone (reference 20 μPa). ABR waveforms were recorded with a Medusa preamplifier at 24.4 kHz and saved to hard disk for offline analysis (Tucker-Davis Technologies). ABR recordings in this study are average waveforms of 300 traces with 10 ms duration. ABR measurements per litter were completed in 30–40 minutes, including the time it took for pups to recover from anesthesia. After recovery, all pups were placed back into their home cages until the next day of testing.

### Combined functional/structural analysis

On the first day of experiments at P10, pups within a litter were labeled with permanent ink, sexed and processed for ABR measurements in pairs. After ABR experiments, anesthetized pups were decapitated and their heads were processed fresh for micro-CT X-ray (micro-CT) imaging. Micro-CT images were acquired and processed as described previously [30]. X-Ray projections were generated around the samples with 0.4˚ rotation steps at a resolution of 11.5 μm per pixel using a 1172 Bruker SkyScan (Bruker). Scans were loaded into MIMICS (v14.0, Materialise) for segmentation and 3D reconstruction. With the exception of the postalignment compensation, all reconstruction parameters were applied identically to all scans. Micro-CT imaging was performed blind to LG group. This procedure was repeated between P10 and P15 until all pups within a litter were used. For the low-LG and the high-LG litters obtained from the last selection experiment, all pups were screened for ABRs between P10 and P15 and pairs were removed daily for micro-CT imaging. This procedure allowed us to track ABRs between P12 and P13, when major changes in air volume of middle ear cavity and physiological responses were observed.

## Gene expression

Quantitative real-time PCR (qRT-PCR) was performed with QuantStudio 7 Flex Real-time qPCR system (Thermofisher), using protocols available at the Advanced Science and Research Center Epigenetic Core Facility. Briefly, primer pairs were obtained from a commercial vendor (Sigma-Aldrich) and primer specificity was tested with adult rat whole-brain cDNA. Total RNA was isolated from a bank of frozen brains kept at -80 ˚C using a RNA isolation kit according to the manufacturer's instructions (Qiagen). Frozen brain samples were thawed and dissected from 5 different regions: cochlear nucleus (CN), pons (ventral brainstem containing the acoustic stria), inferior colliculus (IC), temporal cortex (here referred as auditory cortex, ACX), and occipital cortex (here referred as visual cortex, VCX). The CN and pons were identified visually on the ventral brainstem surface by an experienced subject who used vascular and morphological landmarks as described previously [31]. The IC was visually identified on the dorsal surface as the posterior cuadrant of the midbrain roof, which is well defined in pups of all ages tested. Approximate bregma/lambda coordinates for all dissected brain structures in this study are listed in S1 Table using published criteria [32]. Dissection was performed with autoclaved forceps and scissors, and samples collected in 1 ml tubes filled with chilled RNA free buffer. Reverse transcription and specific target amplification were completed using qScript cDNA Supermix (Quanta) according to manufacturer's protocol. A primer mixture containing both forward and reverse primers was mixed with cDNA from different brain regions and loaded onto 384 well plates. The QuantStudio analysis software was used for data analysis and visualization. Threshold was determined automatically and Ct values were calculated using QuantStudio analysis software.

Table 3 shows the list of 30 primers used in this study. We screened for the expression of genes identified in previous studies of subcortical and cortical brain regions during postnatal development, and focused on genes involved in the hypoxia-sensitive pathway and neurotrophin signaling [33–37]. In this study the housekeeping gene *Actb* (coding for β-actin) was used as a reference for all ages and LG conditions tested. Analysis of the hypoxia-sensitive pathway included genes coding for hypoxia-inducible factors (*Hif1a* and *Epas1*) and prolyl hydroxylases (*Egln1-3*), which are involved in oxygen-sensitive transcriptional regulation [23, 38]. Analysis of neurotrophin signaling included genes coding for brain derived neurotrophic factor (*Bdnf*), nerve growth factor (*Ngf*) and the Bdnf receptor TrkB (*Ntrk2*). Other genes coding for proteins involved in the signaling pathways of interest included transcription factors (*Jun*, *Fos*, and *NFkb1*), kinases (*Akt1*, *Akt2*, and *mTor*), intracellular sorting (*Sort1*), and secreted proteins (*Wnt7a*) [39]. Additional genes involved in cell signaling or commonly used as physiological markers of neurons, vascular and glial cells were screened (*Otx2*, *Olig2*, *Mbp*, *Wnt7b*, *Slc17a8*, *Aqp4*, *Kcna3*, *Ano1*, *Panx1*, *Panx2*, *Gjd2*, *Gja4*, *Gja5*, and *Gja1*). In this study we report relative changes in the mRNA levels of interest as a fold change with respect to birth (P0).

## Data analysis

ABR recordings were saved as text files and analyzed using NeuroMatic in Igor Pro software (WaveMetrics) [40]. ABR thresholds were determined using an amplitude criterion to detect responses that were larger than four times the standard deviation (SD) of the baseline [41]. In general, waveforms with amplitudes larger than 1 microvolt were considered auditory responses. ABR wave 1 was defined as a positive transient voltage change with a peak latency of 1.8 ms to 2.2 ms. Short latency potentials (SLPs) were defined as positive transient voltage changes with a peak latency ~ 1 ms.

**Table 3. List of primer pairs used for analysis of gene expression with qRT-PCR.**

| Gene | Gene ID | Sequence (5'-3') forward; reverse |
|------|---------|-----------------------------------|
| Actb | 81822 | AAGACCTCTATGCCAACAC; TGATCTTCATGGTGCTAGTAGG |
| Hif1a | 29560 | GAAAGGATTACTGAGTTGATGG; CAGACATATCCACCTCTTTTTG |
| Epas1 | 29452 | GATGACAGAATCTTGGAACTG; CACACATATCCTCCATGTTTG |
| Egln1 | 308913 | GAATCAGAACTGGGATGTTAAG; TTGGCATCAAAATACCAGAC |
| Egln2 | 308457 | AAACTCAATTTCATGAGCAGG; CTGAGGTGTTGAACAGAAAC |
| Egln3 | 54702 | TGGGGATCCTAATTATCCAG; TCCTGTCCCTCTCATTTAAC |
| Bdnf | 24225 | GGAGACGAGATTTTAAGACAC; CCATAGTAAGGAAAAGGATGG |
| Ngf | 310738 | AAACTAGGCTCCCTGAAG; AGAACAACATGGACATTACG |
| Ntrk2 | 25054 | AATGGAGACTACACCCTAATG; GAGGGGATCTCATTACTTTTG |
| Akt1 | 24185 | GGGGAATATATTAAAACCTGGC; GTCTTCATCAGCTGACATTG |
| Akt2 | 25233 | GAGTCCTACAGAATACCAGG; AATCTCTGCACCATAAAAGC |
| Sort1 | 83576 | CTTTACCACCCATGTGAATG; TTTTGAAGGTTTTCCCCAAG |
| Wnt7a | 114850 | ATCATCGTCATAGGAGAAGG; ATAATTGCATAGGTGAAGGC |
| Wnt7b | 315196 | CATGAACCTTCACAACAATG; TTGTACTTCTCCTTGAGTAGG |
| Mtor | 56718 | AGAAATTTGATCAGGTGTGC; TTCCTTTTCCTTCTTGACAC |
| Jun | 24516 | AAAAGTGAAAACCTTGAAAGC; CGTGGTTCATGACTTTCTG |
| Fos | 314322 | AAAACTGGAGTTTATTTTGGC; CACAGACATCTCCTCTGG |
| Nfkb1 | 81736 | AAAAACGAGCCTAGAGATTG; ACATCCTCTTCCTTGTCTTC |
| Otx2 | 305858 | GAGAGGACTACTTTCACGAG; CGATTCTTAAACCATACCTGC |
| Olig2 | 304103 | ATCGAATTCACATTCGGAAG; GAAAAAGATCATCGGGTTCTG |
| Mbp | 24547 | GAGAATTAGCATCTGAGAAGG; AAACACATCACTGTCTTCTG |
| Aqp4 | 25293 | GAAAACCACTGGATATATTGGG; CAGAAGACATACTCGTAAAGTG |
| Gjd2 | 50564 | AAATTTGTGACCCATCTCAG; AAACTGTGTTAGGGCTAATG |
| Gja4 | 25655 | AATTTGACCACCGAGGAG; CATACTGCTTCTTGGATGC |
| Gja5 | 50563 | GTGTATATGTGTGTGTGTGC; AGGGCTCTTCTTTACCATTC |
| Gja1 | 24392 | AAAACGTCTGCTATGACAAG; CACAGACACGAATATGATCTG |
| Kcna3 | 29731 | AACTTCAATTACTTCTACCACC; ACTTACTCAGAGTGGAGTTAC |
| Panx1 | 315435 | CTTGACAAAGTCTATAACCGC; ATTAGGTGACTGGAGTTCTTC |
| Panx2 | 362979 | AAACAGCAAGACTGAGAAG; TATAGGGATGCACATCCAAG |
| Slc17a8 | 266767 | CCTGTCTATGCCATTATTGTG; AGAGACCCACCTTACTTATTG |
| Ano1 | 309135 | GAAATCCTGAAGAGAACAACG; TTTACTTAGAAGGGCAGAGTC |

Developmental curves of percent pups with a wave 1 response, EO, or air volume at different ages were fit to Eq 1:

$$Y = Y_0 + (Y_{max} - Y_0)/[1 + \exp(A_{50} - X/k)] \qquad (1)$$

Where $Y_0$ is the minimum observed Y (i.e., the percent pups with ABR, EO, or air volume), Ymax is the maximum observed Y, $A_{50}$ is the age at which Y is half maximum, X is age (in days), and k is the rate coefficient.

## Results

### Variation in maternal licking and grooming (LG)

Fig 2 shows box plots of maternal LG scores from the four cohorts used in this study. The six-day average LG score for each cohort was (mean ± SD): 9.8 ± 1.8 (cohort 1); 7.7 ± 2.2 (cohort

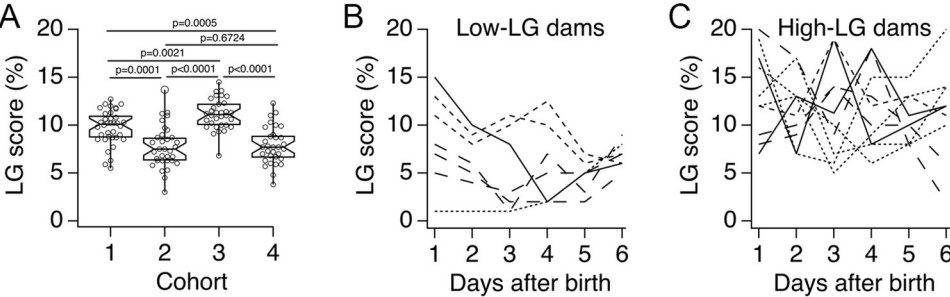

**Fig 2. Variation of licking and grooming (LG) scores in selected dams. A**, Box plots of six-day average LG scores from the four dam cohorts used in this study. Each circle represents one dam. P values were obtained by the Holm-Sidak's multiple comparisons test. **B**, Daily LG scores for seven selected low-LG dams. **C**, Daily LG scores for ten selected high-LG dams. Continuous, dotted, short-dash and long-dash lines in B and C represent dams from cohorts 1, 2, 3 and 4, respectively.

2); 11.2 ± 1.6 (cohort 3); and 7.9 ± 1.9 (cohort 4). Statistical analysis showed significant differences between mean LG scores (Fig 2A; $F_{(3,133)} = 25.91$, $p < 0.0001$, ANOVA). The large variability in LG scores across cohorts prompted us to examine the daily LG scores of the seven low-LG dams and the ten high-LG dams that were selected for developmental tracking experiments. Most low-LG dams showed daily LG profiles that started high and decreased during the six-day observation period (Fig 2B), while high-LG dams had daily LG scores that were very variable but stayed relatively high throughout the six-day observation period (Fig 2C). Overall, these results show the variability of LG scores in selected low-LG and high-LG dams.

## Variation in auditory brainstem response (ABR) onset and eye opening (EO) in pups reared by low-LG and high-LG dams

The ABR and EO were tracked in a total of 81 pups from seven low-LG litters and in 118 pups from ten high-LG litters. For each litter the percent of pups with an ABR wave 1, or the percent of pups with EO were plotted at different ages, and fits to Eq 1 were obtained (Fig 3A–3D). To examine the variation in ABR onset and EO within and across LG groups, the distributions of $A_{50}$ values were compared (Fig 3E). This qualitative analysis showed skewed ABR $A_{50}$ distributions for low-LG and high-LG litters, indicating that the majority of pups examined had early ABR onset between P11.5-P12. However, in some litters pups had ABR onset as late as P13-P13.5. EO $A_{50}$ distributions for low-LG and high-LG litters were also skewed and covered a range of two days between P13 and P15. Statistical analysis showed significant differences between $A_{50}$ medians (Kruskal-Wallis statistic $(4,34) = 24.56$, p value $< 0.0001$, ANOVA). A more detailed examination showed that the ABR $A_{50}$ medians between low-LG and high-LG litters were not significantly different from each other (Fig 3E). A similar result was obtained for EO $A_{50}$ medians (Fig 3E). However, the ABR $A_{50}$ medians were significantly different from the EO $A_{50}$ medians, within and across LG groups (Fig 3E). To obtain information on the synchrony of development within litters, ABR and EO rate coefficient k distributions were compared (see Eq 1 and Fig 3F). The data shows evidence of predominately short values of rate coefficient k for low-LG and high-LG litters. Neither ABR rate coefficient k medians nor EO rate coefficient k medians showed significant differences within and across LG groups (Kruskal-Wallis statistic $(4,34) = 5.890$, p value = 0.1171). In sum, experiments described in Figs 1A, 2 and 3 show that despite significant differences in LG scores between selected dams, ABR onset, timing of EO and synchrony of development do not differ between low-LG and high-LG litters. Instead, there is a range of litter-specific early or late times for ABR onset that

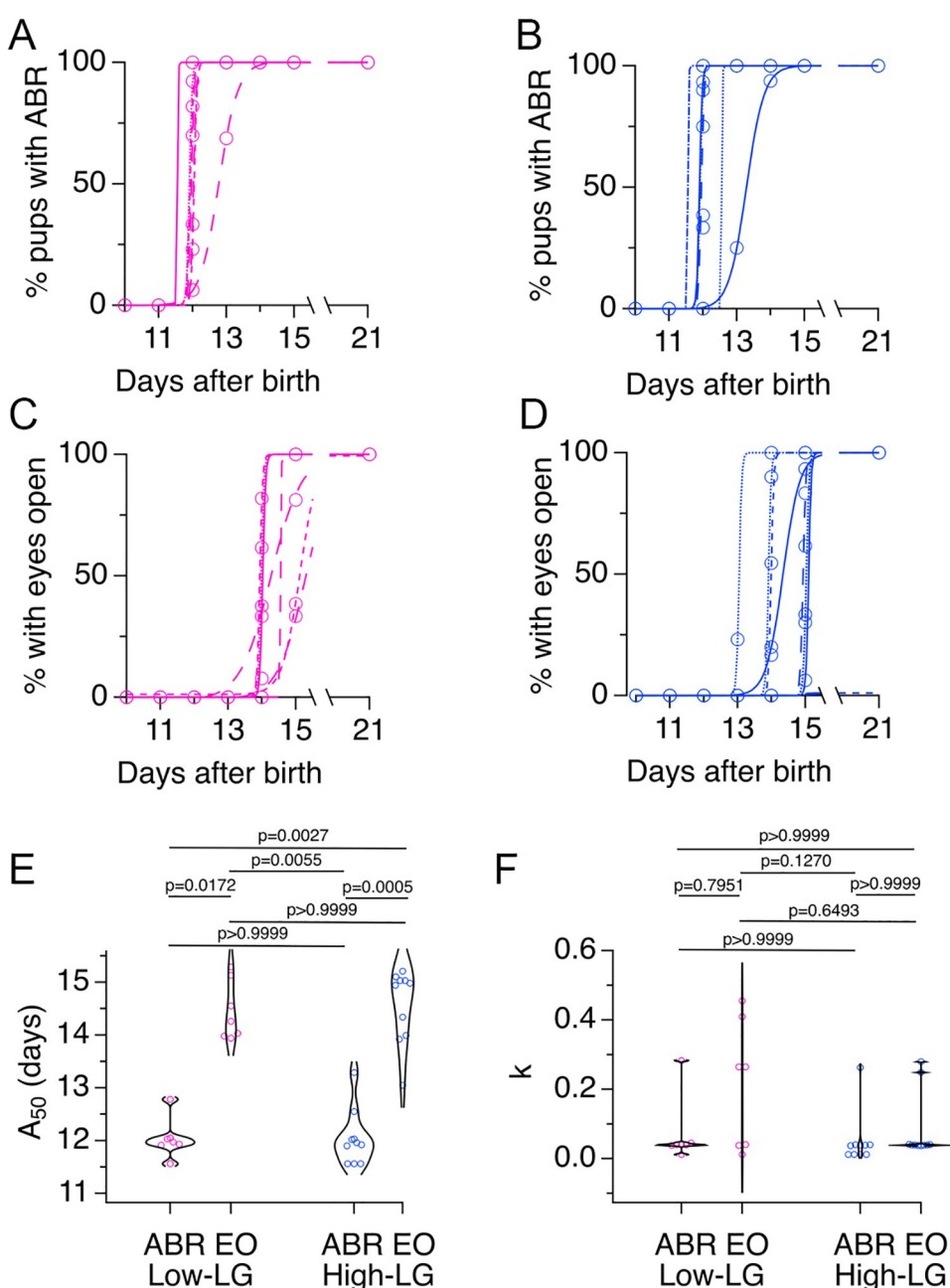

**Fig 3. Timing of auditory brainstem response (ABR) onset and eye opening (EO) in the offspring of selected dams. A**, Plots of percent pups with ABR wave 1 at different ages from seven low licking and grooming (low-LG) litters were fit to Eq 1. **B**, Plots of percent pups with ABR wave 1 at different ages from ten high licking and grooming (high-LG) litters were fit to Eq 1. **C**, Plots of percent pups with EO at different ages from seven low-LG litters were fit to Eq 1. **D**, Plots of percent pups with EO at different ages from ten high-LG litters were fit to Eq 1. **E**, Violin plots of $A_{50}$ values obtained from fits of Eq 1 to developmental data of percent pups with ABR or EO. **F**, Violin plots of rate coefficient k values obtained from fits of Eq 1 to developmental data of percent pups with ABR or EO. P values were obtained by the Kruskal-Wallis multiple comparisons test. Continuous, dotted, short-dash and long-dash lines in A-D represent fits to litter data from cohorts 1, 2, 3 and 4, respectively.

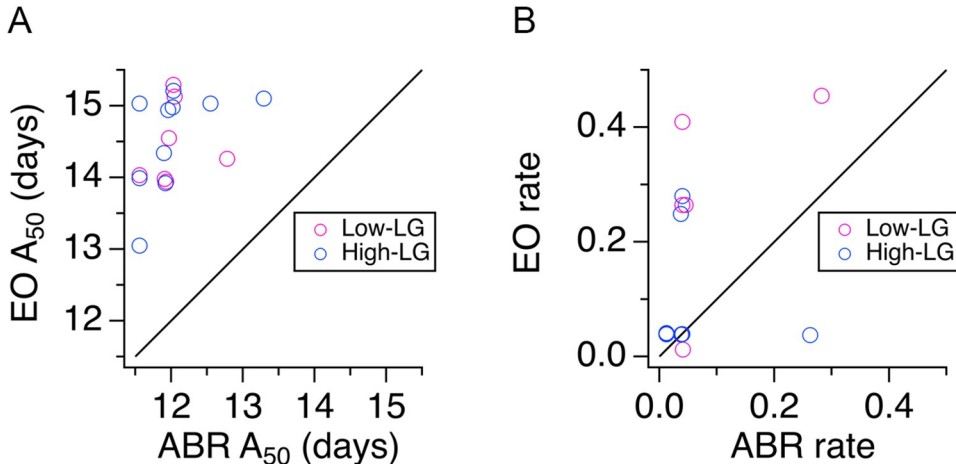

**Fig 4. Delay between auditory brainstem response (ABR) onset and eye opening (EO). A**, Scatter plot of $A_{50}$ values shows the relationship between EO and ABR onset in low licking and grooming (low-LG) and high licking and grooming (high-LG) litters. **B**, Scatter plot of rate coefficient k values shows the relationship between EO and ABR development in selected low-LG and high-LG litters. Magenta symbols represent low-LG data; blue symbols represent high-LG data. Black line in A and B represents the identity line.

happens during a two-day period. Similarly, a two-day range for early and late EO takes place sequentially after ABR onset.

## Differences in the delay between ABR onset and EO in litters with early or late ABR onset

Next, we examined ABR onset and EO data in scatter plots of EO $A_{50}$ values plotted against ABR $A_{50}$ values (Fig 4A). This analysis showed that in individual litters, ABR onset always happened before EO, and notably, it confirmed that for both, ABR and EO, there was a range of early and late onset times that happened within a two-day window: while ABR onset was observed between P11.5 and P13.5, EO was observed between P13 and P15. Note that there was never a litter in which the age of EO coincided with the age of ABR onset. Instead, litters with the earliest ABR onset had more variable EO times than litters with late ABR onset. In litters with ABR onset around P11.5 and P12, we observed delays to EO from 1.5-days to 3.5-days. In contrast, in litters with late ABR onset at ~P13, there was a ~1.5–2 day delay to EO. This pattern was observed in low-LG and high-LG litters alike (Fig 4A). A similar comparison between EO and ABR rate coefficient k showed that in most litters ABR rate coefficient values were <0.1, implying developmental synchrony within litters. In contrast, EO rate coefficients were more variable, implying developmental synchrony and asynchrony, respectively across different litters (Fig 4B).

## Relationship between the development of the middle ear and ABR thresholds in the progeny of low-LG and high-LG dams

To obtain information about developmental structural changes in the auditory periphery of pups from low-LG and high-LG litters, four litters were used to perform correlative ABR and micro CT X-ray tomography (micro-CT) experiments (Fig 1B; n = 51 pups). Detailed examination of the 3D renderings generated from micro-CT data confirmed our previous finding that formation of the middle ear cavity precedes formation of the ear canal (Fig 5A) [30].

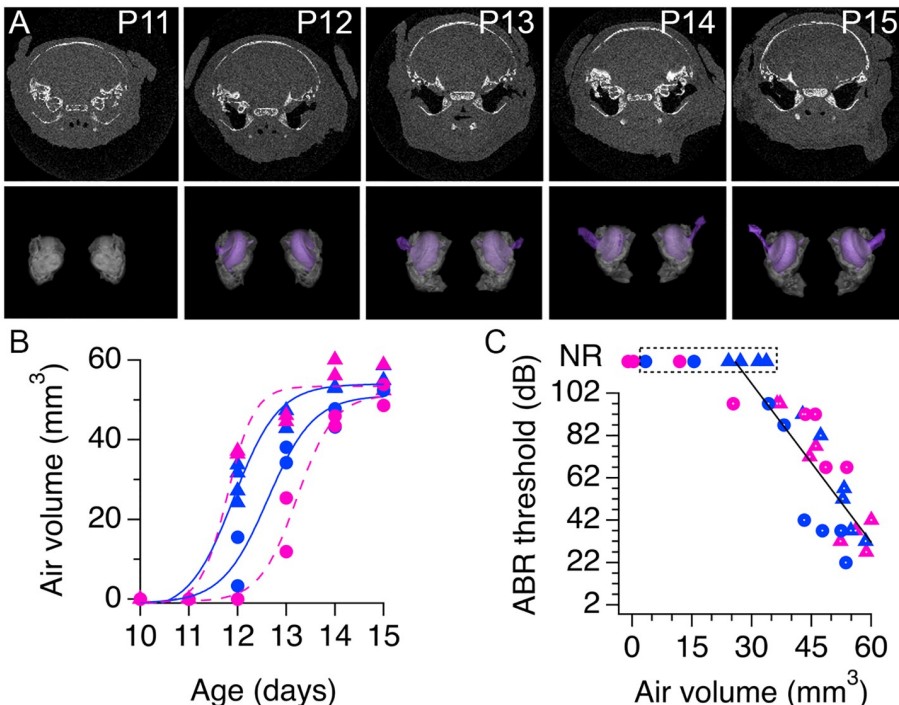

**Fig 5. Relationship between development of the middle ear cavity and auditory brainstem response (ABR) wave 1 thresholds. A**, Top, developmental series of micro-CT X-ray scans of pups from a low licking and grooming (low-LG) litter. White indicates bone, grey is soft tissue and black is air. Bottom, 3D rendering of segmented bone (gray) and air (purple) contrast obtained from tomographic data. **B**, Air volume measured at different ages in pups from two low-LG (magenta) and two high licking and grooming (high-LG) litters (blue). Note that every symbol represents one pup and that similar symbols represent pups from the same litter. Continuous and dotted color lines are fits of Eq 1 to the data. **C**, Relationship between ABR wave 1 thresholds and air volume in the middle ear cavity. Magenta circles and triangles represent pups from low-LG litters (n = 25 pups); blue circles and triangles represent pups from high-LG litters (n = 26 pups). NR = non-responsive pups, defined by the absence of ABR wave 1. Black line represents the fit to a linear function between wave 1 thresholds and air volume with slope 2.5 dB/mm$^3$. Symbols with white dots indicate data points included in the fit.

However, in contrast to previous studies, precursor zones or small air pockets were not observed. Instead, there were marked differences in the air volume of pups from different litters, particularly between P12 and P13 (Fig 5B). Fitting Eq 1 to data in Fig 5B gave $A_{50}$ values that ranged from 11.8 days to 13.2 days (12.4 ± 0.3 days, n = 4 litters), and rate coefficient k values that ranged from 0.28 to 0.47 (0.40 ± 0.04, n = 4 litters). From the 51 pups used in micro-CT imaging experiments, we confirmed an ABR wave 1 in 25 pups between P12 and P15, while 17 pups between ages P10 and P11, and 9 pups between ages P12 and P13 did not show any evidence of ABR wave 1 (Fig 5C). Since the micro-CT imaging did not detect air in any of the 17 non-responsive (NR) pups examined between P10 and P11, we can infer that formation of an air-filled middle ear cavity is necessary for transmission of airborne pressure waves to the inner ear. Linear fitting of wave 1 threshold data versus air volume in the range between 25 mm$^3$ to 60 mm$^3$ gave a slope of -2.5 dB/mm$^3$, showing that auditory thresholds are inversely proportional to air volume in the auditory periphery. However, the structural data also suggests that the presence of air in the middle ear may not be sufficient for proper sound transmission, since there were 7 animals with a measurable air volume in the middle ear at P12 and P13 that did not show an ABR wave 1 (labeled non responsive, NR, in the boxed area of Fig 5C).

To examine the possibility that a minimal air volume at the auditory periphery is necessary for ABR onset, the ABR waveforms from all pups at P12 (n = 10) and all pups at P13 (n = 8) used in the combined ABR and micro-CT experiments were re-examined. To our surprise, seven P12 pups and six P13 pups with air volumes larger than 12 mm$^3$ had responses of comparable amplitude to wave 1, but with a shorter latency. We refer to these events as short latency potentials (SLPs; Fig 6). Fig 6D shows exemplar ABR traces with SLPs at different click intensities in a P12 pup whose structural information is shown in Fig 6C. Note that in this example wave 1 was not present, determined by the absence of a positive potential with a latency ~2 ms. Fig 6F shows exemplar recordings from another P12 pup that had SLPs followed by wave 1 at different click intensities and whose structural data is shown in Fig 6E. Note that in this example a wave 1 was identified after the SLP at click intensities of 102 dB and 97 dB but not at lower intensities, demonstrating that the threshold for the SLP was lower than the threshold for wave 1. Lastly, exemplar ABR traces are shown for a P12 pup with an air volume of zero in the middle ear. In this case SLPs and wave 1 were absent at the same click intensities probed for the other pups (Fig 6A and 6B). Based on these observations, we re-examined all the ABR recordings from the 4 litters used in combined ABR and micro-CT experiments between P11 and P15 to corroborate the presence or absence of SLPs at different ages. We found evidence of SLPs at P12 (n = 7 pups) and P13 (n = 6 pups), but we did not find any evidence of SLPs at P11 (n = 8 pups), P14 (n = 8 pups) and P15 (n = 8 pups). Fig 6G plots the SLP thresholds as a function of air volume for all ten P12 and eight P13 pups used in the combined ABR and micro-CT experiments. Note that there were four NR P12 pups whose air volumes were < 15 mm$^3$ and did not have SLPs or ABR wave 1, and one P13 NR pup with an air volume of 34 mm$^3$ that did not have a SLP but had an ABR wave 1. In Fig 6G, P12 NR pups are enclosed together in a dashed line box, and the P13 NR pup with wave 1 but without SLP is enclosed in a dashed line box marked with an arrow. Linear fitting of SLP threshold data versus air volume in the range between 15 mm$^3$ to 50 mm$^3$ gave a slope of -0.3 dB/mm$^3$. Altogether, these data support the view that a minimal air volume at the auditory periphery is necessary for airborne conduction of click sounds from the external ear to the inner ear. Next, we examined the relationship between SLPs and wave 1 responses.

## SLPs show hallmarks of sensory responses from the inner ear

We hypothesized that SLPs may represent electrical responses in hair cells of the inner ear. Alternatively, SLPs could represent evoked potentials from a different sensory modality, such as somatosensory fibers activated by the pressure energy contained in click stimuli of high intensity. If SLPs were generated in hair cells of the inner ear, then we would expect that SLPs and wave 1 would show hallmarks of synaptic communication, including a defined delay between events. In addition, we would expect developmental changes in the thresholds and the delay between SLPs and wave 1. We would not expect to see these hallmarks if SLPs were sensory responses independent from wave 1. To test these predictions, we took advantage that six littermates from one low-LG litter and six littermates from another high-LG litter were not used for micro-CT scans at P12. First, we recorded ABRs and defined the co-occurrence of SLPs and wave 1. Next, we examined how the threshold for SLPs changed with respect to the threshold of wave 1 between P12 and P13. We found that all six pups from the high-LG litter had SLPs but did not have a wave 1 (open blue triangles in Fig 6H). Interesting to us, five out of six pups from the low-LG litter had SLPs followed by a wave 1, and one pup had SLPs without any evidence of wave 1 (open magenta triangles in Fig 6H). Counting all the pups used in combined ABR and micro-CT experiments and the subset of littermates used in ABR tracking we found that in low-LG litters at P12 there were 2 pups that did not have a SLP nor a wave 1

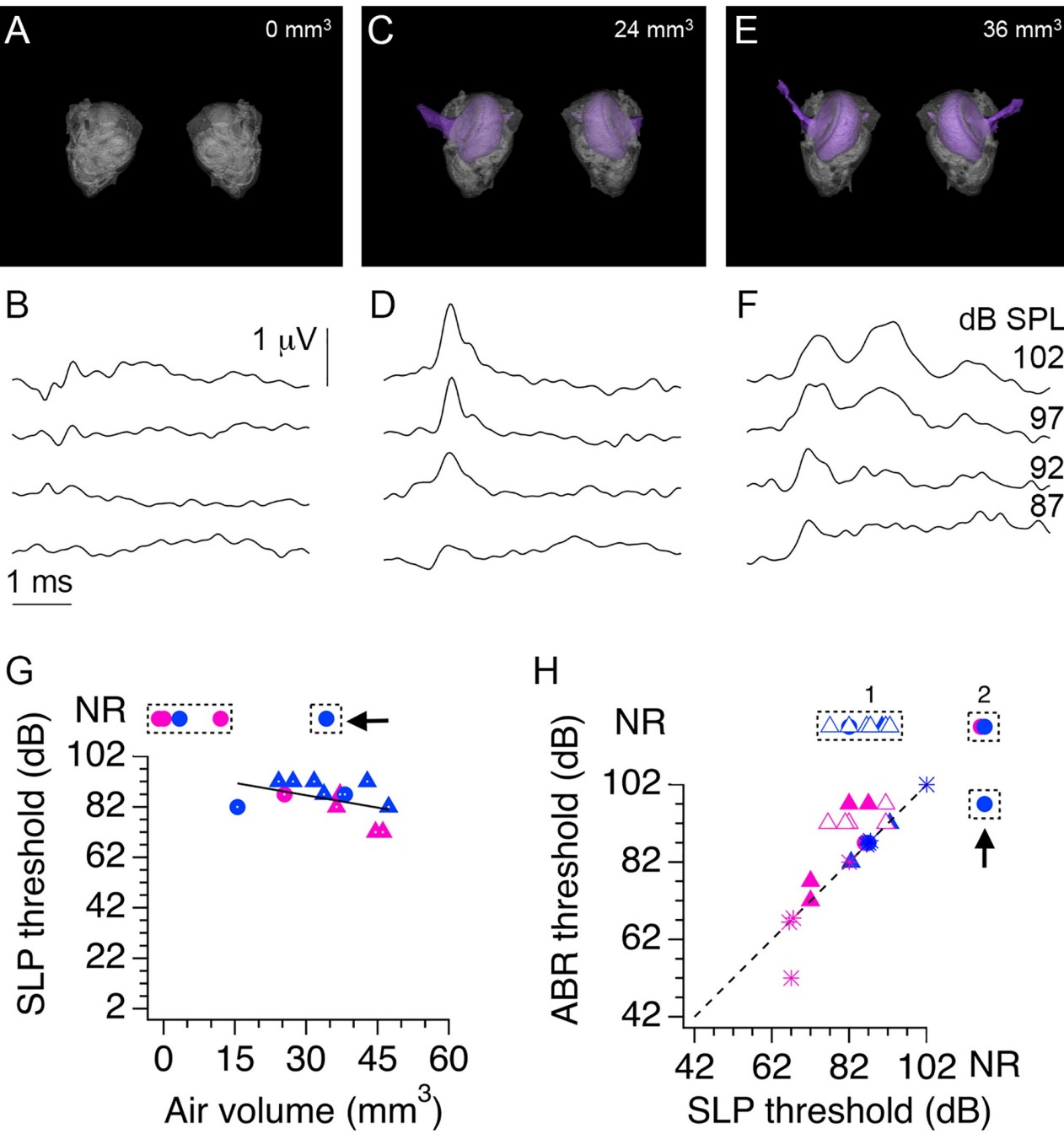

**Fig 6. Identification of short latency potentials (SLPs) in combined auditory brainstem response (ABR) and micro-CT X-ray (micro-CT) experiments. A, C, E**, Representative 3D renderings of three P12 pups with different volumes of air in the middle ear and external canal. **B, D, F**, Corresponding ABR waveforms show the presence of a SLP in pups with air volumes of 24 mm³ and 36 mm³, but not in the pup without middle ear and external ear cavities. ABR traces in B, D and F correspond to click intensities shown in panel F. **G**, SLP thresholds as a function of air volume. Black line represents the fit to a linear function between SLP thresholds and air volume with slope 0.3 dB/mm³. Symbols with white dots indicate data points included in the fit. **H**, Comparison of SLP thresholds and wave 1 thresholds for pups measured at P12 and P13. Filled symbols represent pups used in combined ABR and micro-CT experiments. Open symbols are littermates used solely in ABR experiments at P12. Asterisks indicate pups tracked from P12 to P13. Black dashed line in H is the identity line. Arrows in G and H indicate a pup that showed an ABR wave 1 but not a SLP.

(Fig 6H box 2); 1 pup had SLPs but not a wave 1; 6 pups had SLPs with thresholds that were lower than their corresponding wave 1 thresholds; and 1 pup had SLPs with a threshold that was similar to its wave 1 threshold. In high-LG litters at P12 there was 1 pup without SLP and wave 1; 10 pups had SLPs and no wave 1 (Fig 6H box 1); and 1 pup had SLPs with a threshold lower than its wave 1 threshold. Thus, based on this data, it seems that SLPs occur alone at P12, and when SLPs and wave 1 are observed together, SLPs have lower thresholds than wave 1. In low-LG litters at P13, there were 3 pups that had SLPs with thresholds that were lower than wave 1 thresholds. In high-LG litters at P13, all 3 pups had SLPs with thresholds that were similar to their corresponding wave 1 thresholds. Thus, at P13, SLPs always co-occur with wave 1 and had higher or similar thresholds than wave 1.

To examine the development of SLPs and wave 1 responses, we tracked the ABRs from P12 to P13 in the eight remaining littermates from the low-LG and high-LG litters (4 pups per litter). We found evidence of a decrease in wave 1 thresholds from P12 to P13 such that in 7 of 8 pups SLPs had thresholds that were similar to their corresponding wave 1 thresholds, and in one pup the wave 1 threshold was lower than its corresponding SLP threshold (asterisks in Fig 6H). This single observation raised the possibility that in this animal, SLPs were independent events of wave 1 events. To test the possibility that somatosensory fibers could be activated by the pressure wave energy of high intensity click stimuli, we injected the local anesthetic lidocaine around the skin pad surrounding the pinna area in three P13 littermates used for ABR experiments. This manipulation did not affect the occurrence of SLPs in these animals. Altogether, these results indicate that SLPs antecede the developmental expression of wave 1 responses (open symbols in Fig 6H) and suggest that as pups mature, wave 1 thresholds decrease to match SLP thresholds.

Altogether, data in Figs 5 and 6 show the relationship between development of the auditory periphery and the type of sensorineural response recorded in the progeny of low-LG and high-LG pups. SLPs predominated over wave 1 responses at P12, and gradually waned as wave 1 responses increased in amplitude at P13 and thereafter.

## Analysis of gene expression in the auditory brainstem, inferior colliculus (IC), auditory cortex (ACX) and visual cortex (VCX) of pups reared by low-LG and high-LG dams

After obtaining information on the development of the sensory periphery, we next screened for gene expression changes in the cochlear nucleus (CN), the pons, the IC, the ACX, and the VCX of pups reared by low-LG and high-LG dams at different ages. Table 4 summarizes the results of genes coding for the housekeeping gene *Actb*, and for proteins involved in the hypoxia-sensitive pathway and neurotrophin signaling, expressed as fold-changes with respect to P0. Statistically significant changes between P0 and any age are indicated in bold text. Statistically significant changes between low-LG and high-LG pup samples at a given age are indicated in italics and asterisks. We found statistically significant differences in the expression of *Hif1a* in the pons and ACX of high-LG pups at P21. The expression of *Epas1* also increased significantly at P21, although more consistently, across all examined brain regions of low-LG and high-LG pups. Increased expression of *Epas1* was detected as early as P15 in the CN of high-LG pups, in the IC of low-LG pups, and in the pons, ACX and VCX of low-LG and high-LG pups. As expected from the results on *Hif1a* and *Epas1* expression, we found statistically significant increases in the expression of *Egln* paralogues at P21. For example, *Egln1* increased in the CN, IC, ACX and VCX of low-LG and high-LG pups; *Egln2* increased in the CN of low-LG and high-LG pups; and *Egln3* increased in the CN, pons and VCX of low-LG and high-LG pups, and in the ACX of low-LG pups at P21. Increased expression of *Egln2* was detected as

**Table 4. Fold-change in brain region expression of the housekeeping gene *Actb* and of genes involved in the hypoxia-sensitive pathway and neurotrophin signaling.**

| Gene | Region | P0 | LG | P7 | P15 | P21 | ANOVA |
|------|--------|-----|-----|-----|------|------|-------|
| *Actb* | CN | 1.0±0.08 | Low | 1.0±0.05 | 1.0±0.08 | 1.0±0.03 | $F_{(6,14)} = 0.4522$, p = 0.8318 |
| | | | High | 1.0±0.07 | 1.0±0.04 | 1.0±0.02 | |
| | Pons | 1.0±0.04 | Low | 1.0±0.08 | 1.0±0.03 | 1.1±0.05 | $F_{(6,14)} = 0.6892$, p = 0.6620 |
| | | | High | 1.0±0.01 | 1.0±0.13 | 1.0±0.02 | |
| | IC | 1.0±0.05 | Low | 1.0±0.07 | 1.0±0.02 | 1.1±0.05 | $F_{(6,14)} = 1.093$, p = 0.4129 |
| | | | High | 1.0±0.04 | 1.0±0.03 | 1.0±0.0 | |
| | ACX | 1.0±0.04 | Low | 1.0±0.09 | 1.1±0.04 | 1.1±0.0 | $F_{(6,14)} = 0.6037$, p = 0.7234 |
| | | | High | 1.1±0.04 | 1.0±0.1 | 1.1±0.2 | |
| | VCX | 1.0±0.07 | Low | 1.0±0.06 | 0.9±0.2 | 1.1±0.04 | $F_{(6,14)} = 0.4786$, p = 0.8134 |
| | | | High | 1.0±0.1 | 1.0±0.05 | 1.1±0.05 | |
| *Hif1a* | CN | 1.0±0.4 | Low | 0.6±0.1 | 0.7±0.2 | 1.0 ±0.03 | $F_{(6,14)} = 0.9742$, p = 0.4775 |
| | | | High | 0.8±0.2 | 0.9±0.2 | 1.1±0.01 | |
| | Pons | 1.0±0.01 | Low | 1.2±0.2 | 1.4±0.4 | 1.8±0.07 | $F_{(6,14)} = 1.832$, p = 0.1645 |
| | | | High | 1.2±0.2 | 1.3±0.5 | **2.0±0.07** | |
| | IC | 1.0±0.04 | Low | 0.7±0.2 | 1.1±0.4 | 1.6±0.2 | $F_{(6,14)} = 1.237$, p = 0.3494 |
| | | | High | 0.9±0.2 | 1.0±0.2 | 1.3±0.05 | |
| | ACX | 1.0±0.03 | Low | 1.1±0.2 | 0.6±0.2 | 1.3±0.01 | $F_{(6,14)} = 2.550$, p = 0.0699 |
| | | | High | 0.9±0.2 | 1.0±0.2 | **1.6±0.1** | |
| | VCX | 1.0±0.04 | Low | 0.8±0.3 | 1.0±0.2 | 1.3±0.04 | $F_{(6,14)} = 2.528$, p = 0.0716 |
| | | | High | 0.7±0.3 | 1.2±0.1 | 1.6±0.2 | |
| *Epas1* | CN | 1.0±0.5 | Low | 1.2±0.4 | 1.5±0.4 | **2.6±0.4** | $F_{(6,14)} = 3.918$, p = 0.0165 |
| | | | High | 0.8±0.1 | **2.4±0.6** | **2.6±0.1** | |
| | Pons | 1.0±0.01 | Low | 2.1±0.4 | **3.5±0.7** | **4.7±0.06** | $F_{(6,14)} = 5.757$, p = 0.0033 |
| | | | High | 2.3±0.2 | **3.6±1.5** | **5.4±0.03** | |
| | IC | 1.0±0.1 | Low | 2.1±1.3 | **5.1±2.1** | **4.1±0.3** | $F_{(6,14)} = 3.868$, p = 0.0174 |
| | | | High | 1.5±0.5 | 3.5±0.5 | **6.1±0.4** | |
| | ACX | 1.0±0.6 | Low | 1.3±0.3 | **2.9±0.4** | **6.1±0.8** | $F_{(6,14)} = 13.40$, p<0.0001 |
| | | | High | 1.4±0.2 | **3.7±1.0** | **6.9±0.8** | |
| | VCX | 1.0±0.2 | Low | 2.1±0.8 | **8.1±2.0** | **9.0±0.9** | $F_{(6,14)} = 13.26$, p<0.0001 |
| | | | High | 1.8±0.9 | **8.1±1.6** | **11.0±0.8** | |
| *Egln1* | CN | 1.0±0.4 | Low | **0.3±0.1** | **0.3±0.1** | **1.6±0.1** | $F_{(6,14)} = 15.92$, p<0.0001 |
| | | | High | **0.4±0.1** | **0.3±0.1** | **1.7±0.1** | |
| | Pons | 1.0±0.04 | Low | **0.3±0.1** | 0.4±0.06 | 1.5±0.2 | $F_{(6,14)} = 3.457$, p = 0.0262 |
| | | | High | 0.8±0.5 | 0.5±0.3 | 1.2±0.1 | |
| | IC | 1.0±0.3 | Low | **0.3±0.2** | 0.5±0.2 | **2.3±0.2** | $F_{(6,14)} = 20.98$, p<0.0001 |
| | | | High | **0.4±0.1** | **0.4±0.1** | **2.2±0.1** | |
| | ACX | 1.0±0.2 | Low | **0.5±0.2** | **0.4±0.2** | **2.7±0.1** | $F_{(6,14)} = 57.48$, p<0.0001 |
| | | | High | **0.4±0.1** | **0.4±0.08** | **2.6±0.1** | |
| | VCX | 1.0±0.3 | Low | 0.4±0.3 | 0.5±0.1 | **2.6±0.4** | $F_{(6,14)} = 14.19$, p<0.0001 |
| | | | High | 0.6±0.3 | 0.4±0.1 | **2.7±0.2** | |
| *Egln2* | CN | 1.0±0.3 | Low | 1.4±0.2 | 1.2±0.4 | **2.5±0.6** | $F_{(6,14)} = 1.701$, p = 0.1932 |
| | | | High | 1.8±0.6 | 2.0±0.3 | **2.3±0.4** | |
| | Pons | 1.0±0.1 | Low | 1.3±0.2 | 1.2±0.3 | 1.7±0.7 | $F_{(6,14)} = 0.3384$, p = 0.9052 |
| | | | High | 1.4±0.2 | 1.2±0.5 | 1.3±0.3 | |
| | IC | 1.0±0.3 | Low | 1.0±0.4 | 1.4±0.5 | 1.1±0.3 | $F_{(6,14)} = 0.2842$, p = 0.9349 |
| | | | High | 1.1±0.2 | 1.3±0.3 | 0.8±0.05 | |
| | ACX | 1.0±0.03 | Low | **2.5±0.5** | **2.6±0.7** | 1.5±0.1 | $F_{(6,14)} = 2.263$, p = 0.0976 |
| | | | High | 1.9±0.3 | **2.9±0.7** | 1.8±0.3 | |
| | VCX | 1.0±0.2 | Low | 1.6±0.6 | **2.6±0.6** | 1.9±0.1 | $F_{(6,14)} = 2.685$, p = 0.0599 |
| | | | High | 1.4±0.7 | **3.0±0.2** | 1.5±0.1 | |

*(Continued)*

**Table 4.** (Continued)

| Gene | Region | P0 | LG | P7 | P15 | P21 | ANOVA |
|------|--------|-----|-----|------|------|------|-------|
| Egln3 | CN | 1.0±0.2 | Low | 0.4±0.1 | 0.8±0.3 | **3.7±0.7** | F(6,14) = 12.65, p<0.0001 |
| | | | High | 1.0±0.5 | 0.9±0.2 | **4.5±0.7** | |
| | Pons | 1.0±0.3 | Low | 0.4±0.03 | 1.4±0.8 | **3.6±0.4** | F(6,14) = 12.37, p = 0.0001 |
| | | | High | 0.8±0.05 | 1.1±0.6 | **4.5±0.1** | |
| | IC | 1.0±0.5 | Low | 0.7±0.5 | 0.7±0.3 | 1.5±0.2 | F(6,14) = 3.210, p = 0.0338 |
| | | | High | 0.4±0.1 | 0.5±0.05 | 1.9±0.3 | |
| | ACX | 1.0±0.1 | Low | 0.8±0.1 | 2.2±1.6 | **8.7±5.5** | F(6,14) = 1.665; p = 0.2022 |
| | | | High | 0.9±0.3 | 1.3±0.3 | 3.0±0.3 | |
| | VCX | 1.0±0.06 | Low | 0.6±0.3 | 0.8±0.4 | **2.6±0.1** | F(6,14) = 9.432, p = 0.0003 |
| | | | High | 0.9±0.5 | 0.6±0.2 | **2.5±0.06** | |
| Bdnf | CN | 1.0±0.4 | Low | 0.7±0.2 | 1.5±0.7 | 2.0 ±0.7 | F(6,14) = 0.9133, p = 0.5135 |
| | | | High | 2.9±1.9 | 2.8±0.9 | 2.3±0.3 | |
| | Pons | 1.0±0.4 | Low | 0.8±0.3 | 1.9±0.8 | 0.8±0.1 | F(6,14) = 0.9097, p = 0.5157 |
| | | | High | 1.2±0.2 | 5.7±3.3 | 0.5±0.3 | |
| | IC | 1.0±0.6 | Low | 0.7±0.5 | 0.6±0.2 | 1.5±0.2 | F(6,14) = 1.509, p = 0.2458 |
| | | | High | 0.3±0.06 | 0.8±0.1 | 1.1±0.1 | |
| | ACX | 1.0±0.4 | Low | 1.8±0.8 | 6.8±1.0 | 20.0±10.4 | F(6,14) = 2.512, p = 0.0730 |
| | | | High | 3.6±1.0 | 7.2±2.0 | **14.2±4.2** | |
| | VCX | 1.0±0.7 | Low | 0.6±0.2 | 2.0±0.4 | 5.0±1.6 | F(6,14) = 3.861, p = 0.0175 |
| | | | High | 0.8±0.4 | 2.9±1.1 | **2.9±0.1** | |
| Ngf | CN | 1.0±0.8 | Low | 0.08±0.02 | 0.1±0.08 | 0.6±0.08 | F(6,14) = 1.023, p = 0.4500 |
| | | | High | 0.2±0.06 | 0.3±0.1 | 0.6±0.06 | |
| | Pons | 1.0±0.4 | Low | 0.4±0.2 | 1.2±0.7 | 1.4±0.4 | F(6,14) = 1.075, p = 0.4226 |
| | | | High | 0.4±0.09 | 0.8±0.6 | 0.4±0.09 | |
| | IC | 1.0±0.8 | Low | 0.5±0.3 | 0.04±0.01 | 0.7±0.1 | F(6,14) = 1.340, p = 0.3039 |
| | | | High | 0.02±0.01 | 0.04±0.01 | 0.5±0.08 | |
| | ACX | 1.0±0.5 | Low | 1.0±0.5 | 4.0±1.5 | **19.5±4.5**[*] | F(6,14) = 12.88, p<0.0001 |
| | | | High | 2.0±0.5 | 4.0±0.5 | **13.5±2.0**[*] | |
| | VCX | 1.0±0.0 | Low | 0.3±0.1 | 2.0±0.7 | **11.7±3.0** | F(6,14) = 13.39, p<0.0001 |
| | | | High | 0.7±0.0 | 2.3±0.0 | **6.7±0.0** | |
| Ntrk2 | CN | 1.0±0.5 | Low | 2.4±0.8 | 2.3±0.6 | 1.9±0.2 | F(6,14) = 0.8910, p = 0.5272 |
| | | | High | 2.0±0.2 | 2.3±0.6 | 2.5±0.7 | |
| | Pons | 1.0±0.3 | Low | 1.4±0.5 | 1.7±0.4 | 1.6±0.2 | F(6,14) = 0.8966, p = 0.5238 |
| | | | High | 1.5±0.3 | 1.3±0.4 | 1.9±0.09 | |
| | IC | 1.0±0.1 | Low | 2.3±0.5 | 5.4±1.8 | **3.0±0.8** | F(6,14) = 3.252, p = 0.0324 |
| | | | High | 2.9±0.7 | **3.0±0.3** | 5.2±0.6 | |
| | ACX | 1.0±0.4 | Low | 2.8±1.1 | **4.0±0.8** | **4.6±0.5** | F(6,14) = 8.903, p = 0.0004 |
| | | | High | 2.2±0.6 | **5.3±0.2** | **6.6±0.6** | |
| | VCX | 1.0±0.4 | Low | 2.5±1.1 | **4.9±0.5** | **4.1±0.3** | F(6,14) = 6.234, p = 0.0023 |
| | | | High | 2.1±0.5 | **4.0±0.6** | **4.5±0.2** | |

early as P15 in the ACX and VCX of low-LG and high-LG pups, and as early as P7 in the ACX of low-LG pups. We were surprised to find a statistically significant decrease in the expression level of *Egln1* in the CN and ACX of low-LG and high-LG pups at P15. Furthermore, a statistically significant decrease in the expression level of *Egln1* was confirmed in the CN, the IC and the ACX of low-LG and high-LG pups, and in the pons of low-LG pups at P7.

With respect to genes involved in neurotrophin signaling, the expression profile of *Bdnf* did not show statistically significant changes in brainstem and IC samples from low-LG and high-LG pups at any age tested. In contrast, a statistically significant increase was observed in the ACX and VCX of high-LG pups at P21. The expression profile of *Ngf* did not show changes in brainstem and IC at any age tested, while there was a statistically significant increase in the ACX and VCX of low-LG and high-LG pups at P21. Furthermore, this was the first time that we observed a statistically significant difference between low-LG and high-LG pup samples in the ACX (indicated in italics and an asterisk in Table 4). Lastly, we found a statistically significant increase in *Ntrk2* expression in the ACX and VCX of low-LG and high-LG pups at P15 and P21, but not in the CN and pons at any age tested. In the IC, there was a statistically significant increase in *Ntrk2* levels of high-LG pups at P15, and an increase in low-LG pups at P21.

Next, we examined the expression profile of genes that code for proteins involved at different downstream levels in the signaling pathways of interest. The results are summarized in Table 5. The expression profile of *Akt1* showed a statistically significant increase in the CN of low-LG and high-LG pups at P21, and an increase in the CN of high-LG pups at P15. This was the only example that we found where statistically significant differences between low-LG and high-LG pup samples were evident in the CN (indicated in italics and an asterisk in Table 5). The expression profile of *Akt2* showed a statistically significant increase in the CN, IC, ACX and VCX of low-LG and high-LG pups at P21, and an increase in the ACX of high-LG at P15. The expression profile of *Sort1* showed a statistically significant increase in all brain regions tested at P21. This was the only example where a statistically significant difference was noted between samples from low-LG and high-LG pups in the IC (indicated in italics and an asterisk in Table 5). The increase in *Sort1* expression levels was detected as early as P15 in the CN, the ACX and VCX of low-LG and high-LG pups. The expression profile of *mTor* increased in the IC, ACX and VCX of low-LG and high-LG pups at P21. An increase in *mTor* expression levels was noted as early as P15 in the VCX of high-LG pups. The expression profile of Wnt7a showed an increase in the IC of low-LG pups, and an increase in the ACX of low-LG and high-LG pups at P21.

Table 5 also shows data for transcription factor genes *Jun*, *Fos* and *NFkb1*. At P21, the expression of *Jun* showed an increase in the CN, pons, ACX and VCX of low-LG and high-LG pups. Increased expression of *Jun* was noted as early as P15 in the ACX of low-LG and high-LG pups, and in the pons and VCX of low-LG pups. The expression of *Fos* increased in the CN, ACX and VCX of low-LG and high-LG pups, and in the IC of high-LG at P21. An increase in *Fos* was detected as early as P15 in the ACX of low-LG and high-LG pups, and in the VCX of low-LG pups. Lastly, the expression profile of *NFkb1* was notable since it was the second gene that showed a statistically significant decrease at P7, in this case, in the IC of high-LG pups. *NFkb1* also showed significant increases in the IC of low-LG pups, and in the ACX and VCX of low-LG and high-LG pups at P21. This was the second example where there was a statistically significant difference between low-LG and high-LG samples in the ACX at P21 (indicated in italics and an asterisk in Table 5). An increase in *NFkb1* expression was noted as early as P15 in the ACX and VCX of low-LG pups.

Lastly, we examined the expression profile of genes coding for proteins used as cellular markers of vascular, glial cells and neurons (S2 Table). *Olig2*, which is expressed in cells of the oligodendrocyte lineage, increased in the CN of low-LG and high-LG pups at P21. The myelin basic protein gene *Mbp* increased in the CN of low-LG pups at P21, in the pons of low-LG pups at P15, in the IC of high-LG pups at P21, and in the ACX and VCX of low-LG and high-LG pups at P21. Of note, *Aqp4* a gene coding for a water channel expressed in astrocytes, increased in all brain regions of low-LG and high-LG pups at P15 and P21. *Gjd2*, which codes for neuronal connexin 36, increased in the ACX of high-LG pups at all ages tested, and in the ACX of low-LG pups at P21. *Gja1*, a gene coding for connexin 43 expressed in cells of the

**Table 5. Fold-change in brain region expression of genes involved in downstream signaling.**

| Gene | Region | P0 | LG | P7 | P15 | P21 | ANOVA |
|------|--------|-----|-----|-----|-----|-----|-------|
| Akt1 | CN | 1.0±0.5 | Low | 1.8±0.2 | *1.4±0.3** | **2.0±0.07** | $F(6,14) = 4.379$, p = 0.0107 |
| | | | High | 1.7±0.06 | **2.8±0.3** | **2.1±0.05** | |
| | Pons | 1.0±0.03 | Low | 1.05±0.04 | 1.1±0.2 | 1.1±0.2 | $F(6,14) = 0.7952$, p = 0.589 |
| | | | High | 1.3±0.03 | 4.1±3.3 | 1.2±0.05 | |
| | IC | 1.0±0.09 | Low | 1.0±0.2 | 1.1±0.4 | 1.2±0.02 | $F(6,14) = 0.6139$, p = 0.71611 |
| | | | High | 1.1±0.2 | 1.0±0.2 | 1.4±0.1 | |
| | ACX | 1.0±0.5 | Low | 1.4±0.3 | 0.9±0.4 | 1.5±0.07 | $F(6,14) = 1.460$, p = 0.2611 |
| | | | High | 1.5±0.3 | 0.7±0.3 | 1.7±0.1 | |
| | VCX | 1.0±0.04 | Low | 0.6±0.2 | 0.6±0.2 | 1.0±0.01 | $F(6,14) = 1.814$, p = 0.1680 |
| | | | High | 0.5±0.2 | 0.8±0.2 | 1.0±0.05 | |
| Akt2 | CN | 1.0±0.3 | Low | 1.3±0.2 | 1.3±0.4 | **2.8±0.3** | $F(6,14) = 4.245$, p = 0.0121 |
| | | | High | 1.5±0.4 | 1.6±0.3 | **2.6±0.4** | |
| | Pons | 1.0±0.6 | Low | 0.4±0.05 | 0.5±0.1 | 0.8±0.06 | $F(6,14) = 0.6433$, p = 0.6949 |
| | | | High | 0.6±0.1 | 0.6±0.2 | 0.8±0.04 | |
| | IC | 1.0±0.1 | Low | 1.3±0.5 | 1.8±0.8 | **4.6±0.4** | $F(6,14) = 7.187$, p = 0.0012 |
| | | | High | 1.0±0.3 | 2.2±0.8 | **4.0±0.6** | |
| | ACX | 1.0±0.08 | Low | 1.2±0.2 | 2.1±0.3 | **5.0±0.3** | $F(6,14) = 23.42$, p<0.0001 |
| | | | High | 1.2±0.2 | **2.6±0.9** | **5.6±0.1** | |
| | VCX | 1.0±0.4 | Low | 0.5±0.2 | 1.7±0.6 | **2.0±0.1** | $F(6,14) = 6.086$, p = 0.0026 |
| | | | High | 0.3±0.2 | 1.7±0.4 | **2.6±0.06** | |
| Sort1 | CN | 1.0±0.2 | Low | 1.1 ±0.4 | **2.7±0.2** | **5.5±0.6** | $F(6,14) = 19.18$, p<0.0001 |
| | | | High | 2.1±0.9 | **2.9±0.8** | **6.8±0.6** | |
| | Pons | 1.0±0.3 | Low | 0.3±0.08 | 0.9±0.08 | **1.9±0.5** | $F(6,14) = 14.85$, p<0.0001 |
| | | | High | 0.6±0.1 | 0.7±0.2 | **2.4±0.08** | |
| | IC | 1.0±0.07 | Low | 1.4±0.8 | 3.0±0.5 | *5.5±2.7** | $F(6,14) = 6.416$, p = 0.0020 |
| | | | High | 1.6±0.1 | 1.9±0.6 | *9.6±1.6** | |
| | ACX | 1.0±0.2 | Low | 1.0±0.4 | **4.5±0.9** | **11.5±1.3** | $F(6,14) = 41.38$, p<0.0001 |
| | | | High | 1.3±0.4 | **4.5±0.6** | **12.3±0.9** | |
| | VCX | 1.0±0.2 | Low | 1.0±0.3 | **5.4±0.7** | **12.9±0.5** | $F(6,14) = 40.47$, p<0.0001 |
| | | | High | 1.7±0.6 | **4.2±0.8** | **12.1±1.6** | |
| mTor | CN | 1.0±0.3 | Low | 1.1±0.08 | 0.9±0.2 | 1.5±0.3 | $F(6,14) = 1.401$, p = 0.2815 |
| | | | High | 1.1±0.3 | 1.2±0.2 | 1.7±0.2 | |
| | Pons | 1.0±0.01 | Low | 0.9±0.08 | 0.8±0.2 | 1.0±0.07 | $F(6,14) = 0.5364$, p = 0.7722 |
| | | | High | 1.1±0.3 | 0.8±0.4 | 1.1±0.03 | |
| | IC | 1.0±0.1 | Low | 1.2±0.3 | 1.5±0.6 | **2.1±0.3** | $F(6,14) = 2.148$, p = 0.1119 |
| | | | High | 1.1±0.4 | 1.4±0.3 | **2.3±0.1** | |
| | ACX | 1.0±0.1 | Low | 1.4±0.3 | 1.3±0.5 | **2.2±0.2** | $F(6,14) = 5.120$, p = 0.0056 |
| | | | High | 1.3±0.1 | 1.5±0.3 | **2.7±0.1** | |
| | VCX | 1.0±0.4 | Low | 1.4±0.5 | 1.9±0.5 | **3.3±0.1** | $F(6,14) = 6.824$, p = 0.0015 |
| | | | High | 1.1±0.6 | **2.9±0.4** | **3.5±0.2** | |
| Wnt7a | CN | 1.0±0.6 | Low | 0.3±0.03 | 0.2±0.1 | 1.5±0.6 | $F(6,14) = 2.571$, p = 0.0682 |
| | | | High | 0.6±0.1 | 0.5±0.2 | 1.5±0.2 | |
| | Pons | 1.0±0.4 | Low | 0.3±0.06 | 0.4±0.1 | 0.7±0.1 | $F(6,14) = 0.8058$, p = 0.5819 |
| | | | High | 0.5±0.07 | 1.0±0.8 | 0.8±0.1 | |
| | IC | 1.0±0.2 | Low | 0.9±0.5 | 0.9±0.3 | **2.8±0.5** | $F(6,14) = 4.213$, p = 0.0125 |
| | | | High | 1.0±0.2 | 1.1±0.2 | 1.7±0.1 | |
| | ACX | 1.0±0.3 | Low | 1.3±0.6 | 1.2±0.5 | **2.4±0.1** | $F(6,14) = 4.001$, p = 0.0153 |
| | | | High | 0.9±0.1 | 1.2±0.2 | **2.5±0.3** | |
| | VCX | 1.0±0.2 | Low | 0.6±0.3 | 0.8±0.2 | 2.1±0.2 | $F(6,14) = 1.834$, p = 0.1640 |
| | | | High | 1.8±1.1 | 1.0±0.3 | 2.0±0.02 | |

*(Continued)*

**Table 5.** (Continued)

| Gene | Region | P0 | LG | P7 | P15 | P21 | ANOVA |
|---|---|---|---|---|---|---|---|
| *Jun* | CN | 1.0±0.3 | Low | 1.0±0.07 | 1.5±0.2 | **2.3±0.2** | $F_{(6,14)} = 6.337$, p = 0.0022 |
| | | | High | 1.4±0.3 | 1.5±0.3 | **3.0±0.5** | |
| | Pons | 1.0±0.06 | Low | 1.0±0.2 | 1.7±0.1 | 2.8±0.3 | $F_{(6,14)} = 12.87$, p<0.0001 |
| | | | High | 1.2±0.09 | 1.4±0.2 | 2.5±0.2 | |
| | IC | 1.0±0.2 | Low | 1.1±0.5 | 1.9±0.5 | 3.8±0.5 | $F_{(6,14)} = 10.19$, p = 0.0002 |
| | | | High | 1.0±0.2 | 1.3±0.1 | **3.0±0.1** | |
| | ACX | 1.0±0.05 | Low | 1.2±0.4 | **2.3±0.5** | 3.5±0.5 | $F_{(6,14)} = 13.68$, p<0.0001 |
| | | | High | 1.2±0.2 | **2.6±0.07** | 3.9±0.3 | |
| | VCX | 1.0±0.01 | Low | 1.3±0.5 | **3.8±1.7** | 4.6±0.4 | $F_{(6,14)} = 4.448$, p = 0.0101 |
| | | | High | 1.4±0.4 | 2.1±0.2 | **4.2±0.2** | |
| *Fos* | CN | 1.0±0.3 | Low | 1.0±0.07 | 1.5±0.2 | **2.3±0.2** | $F_{(6,14)} = 5.815$, p = 0.0032 |
| | | | High | 1.4±0.3 | 1.5±0.3 | **3.0±0.5** | |
| | Pons | 1.0±0.1 | Low | 1.0±0.2 | 1.7±0.1 | 2.8±0.4 | $F_{(6,14)} = 1.624$, p = 0.2129 |
| | | | High | 1.2±0.1 | 1.4±0.2 | 2.5±0.2 | |
| | IC | 1.0±0.2 | Low | 1.1±0.5 | 1.9±0.5 | 3.8±0.5 | $F_{(6,14)} = 2.995$, p = 0.0426 |
| | | | High | 1.0±0.2 | 1.3±0.1 | **3.0±0.1** | |
| | ACX | 1.0±0.1 | Low | 1.2±0.4 | **2.3±0.5** | 3.5±0.5 | $F_{(6,14)} = 4.030$, p = 0.0148 |
| | | | High | 1.2±0.2 | **2.6±0.1** | 3.9±0.3 | |
| | VCX | 1.0±0.01 | Low | 1.3±0.5 | **3.8±1.7** | 4.6±0.4 | $F_{(6,14)} = 4.763$, p = 0.0076 |
| | | | High | 1.4±0.4 | 2.1±0.2 | **4.2±0.2** | |
| *Nfkb1* | CN | 1.0±0.2 | Low | 0.4±0.06 | 0.8±0.08 | 1.8±0.6 | $F_{(6,14)} = 1.230$, p = 0.3487 |
| | | | High | 0.6±0.2 | 2.4±1.6 | 1.6±0.2 | |
| | Pons | 1.0±0.2 | Low | 0.4±0.05 | 1.7±0.6 | 3.0±1.1 | $F_{(6,14)} = 1.120$, p = 0.3995 |
| | | | High | 0.6±0.1 | 2.8±2.3 | 1.0±0.1 | |
| | IC | 1.0±0.2 | Low | 0.6±0.2 | 0.9±0.1 | **3.0±0.1** | $F_{(6,14)} = 29.24$, p<0.0001 |
| | | | High | **0.3±0.03** | 0.9±0.2 | 1.2±0.08 | |
| | ACX | 1.0±0.2 | Low | 0.7±0.1 | **2.0±0.1** | **4.6±0.5**[*] | $F_{(6,14)} = 31.90$, p<0.0001 |
| | | | High | 0.9±0.2 | 1.7±0.1 | **2.9±0.05**[*] | |
| | VCX | 1.0±0.05 | Low | 0.6±0.02 | **3.0±0.4** | 3.9±0.6 | $F_{(6,14)} = 21.81$, p<0.0001 |
| | | | High | 0.8±0.04 | 1.8±0.1 | **3.5±0.3** | |

neurovascular unit, increased in the CN and pons of low-LG and high-LG pups at P21, in the IC of high-LG pups at P21, in the ACX of low-LG and high-LG pups at P15 and P21, and in the VCX of low-LG and high-LG pups at P21. *Kcna3*, which codes for a voltage-sensitive potassium channel involved in neurovascular physiology was increased in the ACX of high-LG pups at P21. *Panx2*, a gene coding for Pannexin2 increased in the ACX of high-LG pups at P15, in the ACX of low-LG and high-LG pups at P21, and in the VCX of low-LG and high-LG pups at P15 and P21. *Slc17a8*, which codes for the glutamate vesicular transporter 3, decreased in the pons of low-LG and high-LG pups at P15. We did not find changes in the relative expression levels of *Otx2*, *Wnt7b*, *Gja4*, *Gja5*, *Ano1* and *Panx1* (S3 Table).

## Discussion

In this study, we found that variation in maternal LG behavior does not correlate with systematic differences in the onset of ABRs, the development of the middle ear, nor the timing of EO. Analysis of gene expression in different brain regions of pups reared by low-LG or high-LG dams showed the first evidence that transcription of genes involved in the hypoxia-sensitive

pathway and neurotrophin signaling is regulated during separate time frames in postnatal development. This information is relevant in the context of neonatal encephalopathy, the poorly understood sensitivity of brainstem auditory neurons to oxygen shortage, and existing data on experience-dependent neuroplasticity in the auditory system. Following is a discussion of the merits and limitations of these findings, including considerations for future studies.

## Relationship between development of the auditory periphery and development of sensorineural responses from the inner ear

Combined functional and structural analyses from this study showed that cavity formation in the middle ear correlated with the type of sensorineural responses tracked in animals of different ages. We found that within a range of air volume from 15 mm$^3$ to 40 mm$^3$, the ABR had very elevated click intensity thresholds and relatively simple waveforms (Figs 5 and 6). For example, short latency potentials (SLPs) predominated over wave 1 responses at P12, and SLPs gradually waned as wave 1 responses increased in amplitude at P13 (Fig 6). Our results have confirmed and expanded on the previous findings of Blatchley, Cooper and Coleman, who described similar short latency responses to tone pips in ether-anesthetized P12 Sprague-Dawley rat pups (referred as summating potentials in their Fig 2) [42]. Nevertheless, it is puzzling to us that at P12 SLPs were observed without wave 1 responses, and that when present, wave 1 responses were observed without subsequent ABR waves. Although the functional changes observed in this study are consistent with an important contribution of conductive development to increased sensitivity (Figs 5 and 6) [43, 44], the suppressive effects of isoflurane anesthesia on auditory responses should be considered as a potential confounding factor [45–47]. To address this issue, future studies are needed to define how functional parameters of early sensory responses are affected by the conscious state of the animal. This will require implementation of innovative methods to track structural and functional changes in non-anesthetized pups as they grow during postnatal development.

## Postnatal changes in gene expression of signaling pathways

The first postnatal weeks represent a sequence of sensitive periods when expression of genes involved in cellular proliferation, migration, differentiation, synaptogenesis, myelination, apoptosis, and survival are regulated temporally and regionally throughout the nervous system. In this study we found that *Egln1* which codes for Phd2, one of the negative regulators of Hif degradation by the ubiquitin-proteasome pathway [23, 38], was down regulated between P0 and P7 in all auditory brain regions examined, but not in the primary visual cortex of low-LG pups. Although there were no significant changes in the pons between P0 and P7, a similar profile of transient decrease in *Egln1* expression was observed in most auditory brain regions, but not in the primary visual cortex of high-LG pups (Table 4). Assuming that transcriptional down regulation of *Egln1* causes reduced levels of mRNA transcripts and Phd2 in cells of the auditory system, this result would suggest that cells in the auditory system of neonate rats may have a differential sensitivity to oxygen challenges during postnatal development. This interpretation is consistent with results that showed higher vulnerability of P7 compared to P1 rats when exposed to hypoxia-ischemia, and with studies that tracked long-term maternal behavioral, learning and auditory deficits in rats exposed to perinatal anoxia at birth [16, 48, 49]. The first postnatal week in the rat auditory system represents a sensitive period for cell proliferation and for the development of a highly demanding metabolic profile [50–52]. Since *Hif1a* and *Epas1* code for proteins that regulate the expression of multiple genes involved in glycolysis and angiogenesis, it can be predicted that changes in the expression of enzymes, transporters and pro-angiogenic factors such as VEGF, could begin before hearing onset, as has been

documented in the superior olivary complex of gerbils for metabolic markers [53]. Our preliminary studies in the auditory brainstem of rats suggest that vascular volume increases between P5 and P10 [54]. Lastly, it is important to note that the hypoxia-sensitive pathway regulates gene expression by a negative feedback mechanism sensitive to a reduction in $O_2$ partial pressure. This opens new questions about the nature of factors that might regulate the transcription of *Egln1* in the central auditory system.

Neurotrophin signaling is involved in neuronal survival, cell growth, and differentiation via activation of tyrosine kinase receptors such as Ntrk2, which in turn can modulate several signaling pathways including Akt/PI3K, Jak/STAT, NF-κB, UPAR/UPA, Wnt/β-catenin, and VEGF [39]. Although the observation that *Akt1*, *Jun*, and *Sort1* are upregulated in auditory brainstem structures is relatively new, in this study we also found that *Akt2*, *Sort1*, *mTor*, *Wnt7a*, *Jun*, and *Nfkb1* were upregulated predominantly in sensory cortex at P15 and P21, which are ages posterior to the onset of ABRs and EO. This result is consistent with studies in rats and a mouse model of fragile X that have shown that Bdnf signaling is regulated by sensory experience [55–58]. In addition, we found evidence that the relative levels of expression of *Ngf*, *Akt1*, *Sort1 and Nfkb1* were significantly different in samples from low-LG and high-LG pups (Tables 4 and 5). These results expand previous findings that maternal care strongly modulates brain levels of the neurotrophin Bdnf in rodents [59, 60], and implicate maternal LG in experience-dependent development of functional responses in primary auditory cortex [61].

## Variable delay between hearing onset and EO

A major caveat of the approach used in the present study is that we were not able to predict the developmental profile of individual litters based on maternal LG. However, we were surprised to find that litters reared by low-LG or high-LG dams showed a similar range of early and late ABR onset. This turned out to be an advantage, because by tracking pups during development we were able to measure the delay between ABR onset and EO in a relatively large number of litters. The onset of hearing for airborne sounds precedes EO, and this sequence of events occurs prenatally or postnatally in different vertebrate species. The finding that litters with an early ABR onset have a more variable delay to EO (Fig 4) is relevant in the context of recent studies that manipulated the timing of EO and measured its effects on the development of membrane and synaptic properties of primary auditory cortex neurons in gerbils, and of synaptic properties of primary visual cortex neurons in Long-Evans rats [62, 63]. We propose that a better understanding of the relationship between hearing onset and the delay to EO will be useful to study cross modal experience-dependent plasticity between visual and auditory systems in rodents. Studies are needed to characterize the signaling pathways that influence the development of feed-forward and feedback mechanisms between ACX and VCX in animals with early and late hearing onset [62, 64, 65].

In conclusion, maternal LG does not seem to affect the development of accessory structures in the auditory periphery. The results of a gene expression screen suggest that sensitivity to hypoxic challenge might be widespread in the auditory system of neonate rats before hearing onset, and that maternal LG may affect transcriptional regulation of genes involved in neurotrophin signaling, which implicates maternal LG behavior in experience-dependent neuroplasticity.

## Supporting information

**S1 Table. Bregma/Lambda reference coordinates (in mm) used for dissection of brain regions in this study.**
(DOCX)

**S2 Table. Fold-change in brain region expression of genes expressed in vascular, glial cells and neurons.**
(DOCX)

**S3 Table. Genes that did not show changes in this study.**
(DOCX)

**S1 Data.**
(XLSX)

**S2 Data.**
(XLSX)

**S3 Data.**
(XLSX)

**S4 Data.**
(XLSX)

**S5 Data.**
(XLSX)

## Acknowledgments

We would like to thank former lab members and students from Macaulay Honors College Seminar 3 for discussions and help with behavioral scoring. Gene expression data was obtained and processed with help from the CUNY-ASRC Epigenetic Core facility staff.

## Author Contributions

**Conceptualization:** Luis Cardoso, Adrián Rodríguez-Contreras.

**Data curation:** Geng Pan, Adrián Rodríguez-Contreras.

**Formal analysis:** Jingyun Qiu, Preethi Singh, Geng Pan, Annalisa de Paolis, Adrián Rodríguez-Contreras.

**Funding acquisition:** Jia Liu, Adrián Rodríguez-Contreras.

**Investigation:** Jingyun Qiu, Preethi Singh, Geng Pan, Annalisa de Paolis, Adrián Rodríguez-Contreras.

**Methodology:** Frances A. Champagne.

**Resources:** Jia Liu, Luis Cardoso.

**Supervision:** Frances A. Champagne, Jia Liu, Luis Cardoso, Adrián Rodríguez-Contreras.

**Visualization:** Annalisa de Paolis.

**Writing – original draft:** Jingyun Qiu, Preethi Singh, Adrián Rodríguez-Contreras.

**Writing – review & editing:** Jingyun Qiu, Preethi Singh, Geng Pan, Annalisa de Paolis, Frances A. Champagne, Jia Liu, Luis Cardoso, Adrián Rodríguez-Contreras.

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
