## [Decision Letter · Decision Letter 0]

27 Apr 2020

PONE-D-20-07005

Robust range of auditory periphery development, eye opening, and brain gene expression in Wistar rat pups that experience variation in maternal backgrounds.

PLOS ONE

Dear Dr. Rodriguez-Contreras,

Thank you for submitting your manuscript to PLOS ONE. After careful consideration, we feel that it has merit but does not fully meet PLOS ONE’s publication criteria as it currently stands. Therefore, we invite you to submit a revised version of the manuscript that addresses the points raised during the review process.

We would appreciate receiving your revised manuscript by Jun 11 2020 11:59PM. To enhance the reproducibility of your results, we recommend that if applicable you deposit your laboratory protocols in protocols.io, where a protocol can be assigned its own identifier (DOI) such that it can be cited independently in the future. For instructions see: http://journals.plos.org/plosone/s/submission-guidelines#loc-laboratory-protocols

We look forward to receiving your revised manuscript.

Kind regards,

Giuseppe Biagini, MD

Academic Editor

PLOS ONE

Journal Requirements:

"IACUC Approval number 1000 Isoflurane anesthesia was used for electrophysiology experiments to minimize pain and prevent animal movement."  

a.) Please amend your current ethics statement to include the full name of the ethics committee/institutional review board(s) that approved your specific study.

b.) Please amend your current ethics statement to confirm that your named institutional review board or ethics committee specifically approved this study.

Additional Editor Comments (if provided):

Reviewers' comments:

Reviewer's Responses to Questions

**Comments to the Author**

1. Is the manuscript technically sound, and do the data support the conclusions?

Reviewer #1: Partly

Reviewer #2: Partly

2. Has the statistical analysis been performed appropriately and rigorously? 

Reviewer #1: Yes

Reviewer #2: Yes

3. Have the authors made all data underlying the findings in their manuscript fully available?

Reviewer #1: Yes

Reviewer #2: Yes

4. Is the manuscript presented in an intelligible fashion and written in standard English?

Reviewer #1: Yes

Reviewer #2: Yes

5. Review Comments to the Author

Reviewer #1: This manuscript details effects of maternal licking and grooming behavior on the neurodevelopment of pups. The authors have mainly focused on the effects of maternal behavior on the development of auditory and visual cortices and respective outcomes. The wide range of testing paradigm was used such as analysis of ABR, micro-CT, time onset of eye opening and gene expression analysis of various pathways involved in growth, and related pathways. The authors found that maternal behavior does not significantly impact ABR onset. The observations indicated a delay between hearing onset and eye opening. The study provides novel indication about relationship between growth factor bdnf RNA expression and hypoxia-related pathway in the LG group.

Comments to the authors:

The study is very elaborate and robust methodology was followed. However, I have a few specific comments

Methods:

1. Page 5 line 109 to 121: The description of animals and the format of testing with numbers followed is elusive, it would be great if all those numbers that were used could be presented in a table format.

2. Page 5 line 113: It is not mentioned where did these 160 females come from?

3. Page 5 line 25 : The description of text does not match that of figures

4. Page 7 line 164: what is the rationale for using median as a statistical entity here

5. Page 9 line 211: Were the observations made manually or using any behavioral software, if manually were they done by the same individual or different? Because the observations can vary with individual perspectives, how was the variation controlled for?

6. Page 11 line 273: What are the indications of looking for these genes, please provide some background while introducing these?

7. Why were the litters selected based on summer and spring, what is the significance of seasons, what is the rationale behind this?

8. Did the authors take into consideration the effect of isoflurane on latency to onset of ABR?

Figures and results and discussion

1. Page 5 line 125 : The figure 1D is not present, which figure are you referring to?

2. Why are the figure legends for figure 7, 8 appearing before figure 4,5,6 ? I think they should be placed well in sequence or the text should be adjusted accordingly.

3. Page 16 line 372: Which figure are you talking about here, please mention in the text

4. How do the findings in the gene expression analysis correlate and reflect upon the ABR, eye opening or other changes in the pups of High and low LG dams, what are author’s insight about the changes at transcription level? Please include in discussion.

5. Is it possible to represent the results of gene expression analysis in tabular form and highlighting the significant ones, it will make the manuscript for comprehensive?

6. Page 37 line 897: When the authors mention about this correlation, are they sure of a direct correlation or there are confounders due to variability in measurements by individuals?

General comments:

The manuscript is thoroughly written and grammatically sound , I just have a few general comments

1. About the title, would you be more specific when you say, ‘maternal background’, specifying that it is the behavioral background you are focusing on rather than the genetic?

2. Moving a lot of gene analysis data into tabular format will help decrease confusion to the reader.

3. Please check and match figures to the figure legends.

Reviewer #2: Review on PONE-D-20-07005:

The authors investigated the effect of maternal licking and grooming (LG) during the first week of life on the timing of hearing onset in Wistar rat pups. In this regard, they examined some parameters such as auditory brainstem responses (ABR), tracking of eye opening (EO), micro-CT X-ray tomography, and qRT-PCR to monitor neurodevelopmental changes in the female and male offspring exposed to different maternal LG. The authors found no significant effect of LG on the mentioned parameters but found that the hypoxia-sensitive pathway is regulated in subcortical and cortical auditory brain regions before hearing onset, and also the role of maternal LG in regulation of BDNF signaling in auditory cortex after hearing onset. However some major points have not been properly addressed as follows:

Title: The title is not in accordance with the findings of the study.

Abstract: Lines 32-34 of this section, are not correlated with the main findings of the research.

Introduction: According to the main findings of the research it is suggested to re-write the introduction in order to get the reader aware of what is presented in the following parts of the manuscript. For example it is needed to be written something about hypoxia-sensitive pathway and BDNF signaling in relation to hearing onset. Also, the importance of the research and the possible application and benefit of the obtained results should be mentioned.

Discussion: As mentioned above in the comments for introduction, the first paragraph of the discussion also needs revision. The second and third parts of the discussion is better to be presented before the first part.

-The conclusion needs revision, it should contain a take-home message actually according to the main findings of the study, not again discussing on other studies.

-As a whole, the manuscript should be integrated and shortened.

6. PLOS authors have the option to publish the peer review history of their article (what does this mean?). If published, this will include your full peer review and any attached files.

Reviewer #1: No

Reviewer #2: No

---

## [Author Response · Author response to Decision Letter 0]

16 Jun 2020

PONE-D-20-07005

Robust range of auditory periphery development, eye opening, and brain gene expression in Wistar rat pups that experience variation in maternal backgrounds.

PLOS ONE

Dear Dr. Biagini,

We would like to thank the reviewers for their constructive reviews and comments. To comply with journal requirements we have updated the ethics statement in the Methods section as follows: ‘The Institutional Animal Care and Use Committee of the City College of New York specifically reviewed and approved this study.’ We have created 2 supplementary files with Tables. We have updated the Data Availability statement appropriately and provide excel files with the data used to make the figures and tables of the manuscript. Next is our point by point response to the comments made by two reviewers:

Reviewers' comments:

Reviewer's Responses to Questions

Comments to the Author

1. Is the manuscript technically sound, and do the data support the conclusions?

Reviewer #1: Partly

Reviewer #2: Partly

2. Has the statistical analysis been performed appropriately and rigorously? 

Reviewer #1: Yes

Reviewer #2: Yes

3. Have the authors made all data underlying the findings in their manuscript fully available?

Reviewer #1: Yes

Reviewer #2: Yes

4. Is the manuscript presented in an intelligible fashion and written in standard English?

Reviewer #1: Yes

Reviewer #2: Yes 

5. Review Comments to the Author

Reviewer #1: This manuscript details effects of maternal licking and grooming behavior on the neurodevelopment of pups. The authors have mainly focused on the effects of maternal behavior on the development of auditory and visual cortices and respective outcomes. The wide range of testing paradigm was used such as analysis of ABR, micro-CT, time onset of eye opening and gene expression analysis of various pathways involved in growth, and related pathways. The authors found that maternal behavior does not significantly impact ABR onset. The observations indicated a delay between hearing onset and eye opening. The study provides novel indication about relationship between growth factor bdnf RNA expression and hypoxia-related pathway in the LG group.

Comments to the authors:

The study is very elaborate and robust methodology was followed. However, I have a few specific comments

Methods:

1. Page 5 line 109 to 121: The description of animals and the format of testing with numbers followed is elusive, it would be great if all those numbers that were used could be presented in a table format.

Authors response: Thank you for your suggestion. We revised and summarized adult animal information in Table 1 and pup information in Table 2. We edited the text on the first paragraph of “Experimental Design and Statistical Analysis” to account for this change.

2. Page 5 line 113: It is not mentioned where did these 160 females come from?

Authors response: This information is now found in Table 1.

3. Page 5 line 25 : The description of text does not match that of figures

Authors response: Thank you for pointing this out. We have revised the manuscript so that text matches the figures.

4. Page 7 line 164: what is the rationale for using median as a statistical entity here

Authors response: The coefficient k value data did not pass a normality test, so we used a non-parametric test to assess statistical differences between the medians of the data samples. We have updated the text in “Experimental Design and Statistical Analysis” to indicate when means or medians were used in statistical tests. 

5. Page 9 line 211: Were the observations made manually or using any behavioral software, if manually were they done by the same individual or different? Because the observations can vary with individual perspectives, how was the variation controlled for?

Authors response: Thank you for your questions. We did not perform software analysis of behavioral video recordings. Instead, lead authors JQ and PS were trained to manually score maternal behaviors by senior author FC. Given the overwhelmingly large number of observations to complete in this study, we decided to recruit different groups of undergraduate students who helped us score maternal behavior. Motivated undergraduate student helpers were recruited by AR-C and were trained by JQ and PS. To improve training and assess scoring variation amongst student helpers, JQ and PS chaperoned each group of students for their first set of two or three observations. We estimate that variation across observers was smaller than 5%. For clarity, we have added text that indicates observations were performed by trained observers.

6. Page 11 line 273: What are the indications of looking for these genes, please provide some background while introducing these?

Authors response: Thank you for your suggestion. This is a point that overlaps with the comments of reviewer 2 below. We have made three changes to the manuscript. We rewrote the introduction to include a rationale for including the hypoxia-sensitive pathway and neurotrophin signaling in the gene expression screen. We provide more background in the Gene Expression section of the Methods to reference previous studies that identified the genes of interest. Lastly, we reorganized the presentation of gene expression data in two new tables (Table 4 and Table 5), and added two supplementary tables for data that is less relevant to the main focus of the paper (S1 Table and S2 Table).

7. Why were the litters selected based on summer and spring, what is the significance of seasons, what is the rationale behind this?

Authors response: Thank you for your question. The main reason to conduct observations in the spring and summer was the availability of space at the animal facility. We did not intend to study seasonal effects in this study, so we have removed any reference to seasons from the manuscript. As a result, we edited Panel A in Figure 2, eliminated panel D in Figure 2, an updated the text of the legend to Figure 3 and throughout the manuscript.

8. Did the authors take into consideration the effect of isoflurane on latency to onset of ABR?

Authors response: Thank you for your observation. We consider that anesthesia could be a confounding factor since it has suppressive effects on neural activity. We added a statement and references to support it in the discussion.

Figures and results and discussion

1. Page 5 line 125 : The figure 1D is not present, which figure are you referring to?

Authors response: Thank you for pointing this out. We have revised the text to match the figures throughout the manuscript.

2. Why are the figure legends for figure 7, 8 appearing before figure 4,5,6 ? I think they should be placed well in sequence or the text should be adjusted accordingly.

Authors response: Thank you for your suggestion. We have revised the text so that all the figure legends appear in the appropriate numerical sequence. 

3. Page 16 line 372: Which figure are you talking about here, please mention in the text

Authors response: We did not generate a figure for the correlation analysis. After revising the manuscript we do not consider this analysis to be essential for the evaluation of the data. In the interest of integrating and shortening the manuscript (as suggested by reviewer 2) we removed this paragraph from the manuscript

4. How do the findings in the gene expression analysis correlate and reflect upon the ABR, eye opening or other changes in the pups of High and low LG dams, what are author’s insight about the changes at transcription level? Please include in discussion.

Authors response: We re-organized the discussion section to provide an interpretation of the gene expression data.

5. Is it possible to represent the results of gene expression analysis in tabular form and highlighting the significant ones, it will make the manuscript for comprehensive?

Authors response: Thank you for the suggestion. We replaced Figures 7, 8 and 9 with tables 4 and 5 which focus on the hypoxia-sensitive pathway and neurotrophin signaling, and genes related to these pathways. We generated two supplementary tables with results that are less relevant to the main focus of the paper. 

6. Page 37 line 897: When the authors mention about this correlation, are they sure of a direct correlation or there are confounders due to variability in measurements by individuals?

Authors response: Thank you for pointing this out. It is entirely possible that due to variation of observations by human observers we did not find a correlation between these variables. As indicated in our reply to point 3, we have removed this paragraph from the manuscript.

General comments:

The manuscript is thoroughly written and grammatically sound, I just have a few general comments

1. About the title, would you be more specific when you say, ‘maternal background’, specifying that it is the behavioral background you are focusing on rather than the genetic?

Authors response: Thank you for your suggestion. We rewrote the Title to better reflect the aims and results of the study, and to indicate we used behavioral background as you suggest. As appropriate, we have indicated throughout the manuscript that we refer to LG as a maternal behavior and not a genetic trait.

2. Moving a lot of gene analysis data into tabular format will help decrease confusion to the reader.

Authors response: Thank you for your suggestion. We have simplified gene expression data by focusing our attention in fewer genes listed in Tables 4 and 5, and by creating two supplementary tables.

3. Please check and match figures to the figure legends.

Authors response: Thank you for your suggestion. We checked that figure legends match the content of the corresponding figures.

Reviewer #2: Review on PONE-D-20-07005:

The authors investigated the effect of maternal licking and grooming (LG) during the first week of life on the timing of hearing onset in Wistar rat pups. In this regard, they examined some parameters such as auditory brainstem responses (ABR), tracking of eye opening (EO), micro-CT X-ray tomography, and qRT-PCR to monitor neurodevelopmental changes in the female and male offspring exposed to different maternal LG. The authors found no significant effect of LG on the mentioned parameters but found that the hypoxia-sensitive pathway is regulated in subcortical and cortical auditory brain regions before hearing onset, and also the role of maternal LG in regulation of BDNF signaling in auditory cortex after hearing onset. However some major points have not been properly addressed as follows:

Title: The title is not in accordance with the findings of the study.

Abstract: Lines 32-34 of this section, are not correlated with the main findings of the research.

Authors response: Thank you for pointing this out. We have removed the referred text from the abstract. The abstract has been edited to reflect the main goals and findings of the study.

Introduction: According to the main findings of the research it is suggested to re-write the introduction in order to get the reader aware of what is presented in the following parts of the manuscript. For example it is needed to be written something about hypoxia-sensitive pathway and BDNF signaling in relation to hearing onset. Also, the importance of the research and the possible application and benefit of the obtained results should be mentioned.

Authors response: Thank you for making this suggestion. We have re-written the second and third paragraphs of the introduction to highlight the relevance of the research on hypoxia-sensitive and neurotrophin pathways during auditory system development, including statements of significance and possible clinical applications of the research.

Discussion: As mentioned above in the comments for introduction, the first paragraph of the discussion also needs revision. The second and third parts of the discussion is better to be presented before the first part.

Authors response: We have re-organized the discussion section as suggested.

-The conclusion needs revision, it should contain a take-home message actually according to the main findings of the study, not again discussing on other studies.

-As a whole, the manuscript should be integrated and shortened.

 Authors response: We have revised the manuscript trying to integrate and shorten the text. We have tried to focus our discussion on the main findings of the study and less on results from other studies.

6. PLOS authors have the option to publish the peer review history of their article (what does this mean?). If published, this will include your full peer review and any attached files.

Do you want your identity to be public for this peer review? For information about this choice, including consent withdrawal, please see our Privacy Policy.

Reviewer #1: No

Reviewer #2: No

---

## [Decision Letter · Decision Letter 1]

20 Jul 2020

PONE-D-20-07005R1

PONE-D-20-07005

Defining the relationship between maternal care behavior and sensory development in Wistar rats: auditory periphery development, eye opening and brain gene expression

PLOS ONE

Dear Dr. Rodriguez-Contreras,

Thank you for submitting your manuscript to PLOS ONE. After careful consideration, we feel that it has merit but does not fully meet PLOS ONE’s publication criteria as it currently stands. Therefore, we invite you to submit a revised version of the manuscript that addresses the points raised during the review process.

We look forward to receiving your revised manuscript.

Kind regards,

Giuseppe Biagini, MD

Academic Editor

PLOS ONE

Reviewers' comments:

Reviewer's Responses to Questions

**Comments to the Author**

1. If the authors have adequately addressed your comments raised in a previous round of review and you feel that this manuscript is now acceptable for publication, you may indicate that here to bypass the “Comments to the Author” section, enter your conflict of interest statement in the “Confidential to Editor” section, and submit your "Accept" recommendation.

Reviewer #1: All comments have been addressed

Reviewer #3: All comments have been addressed

2. Is the manuscript technically sound, and do the data support the conclusions?

Reviewer #1: Yes

Reviewer #3: Yes

3. Has the statistical analysis been performed appropriately and rigorously? 

Reviewer #1: Yes

Reviewer #3: Yes

4. Have the authors made all data underlying the findings in their manuscript fully available?

Reviewer #1: Yes

Reviewer #3: Yes

5. Is the manuscript presented in an intelligible fashion and written in standard English?

Reviewer #1: Yes

Reviewer #3: Yes

6. Review Comments to the Author

Reviewer #1: The authors have addressed all the major concerns. And the manuscript is acceptable in its current form. However, I have a few minor comments as follows:

1. Page 3 line 62: Is there a proof of maternal LG behavior leading to hypoxia or oxygen deficiency in pups, please mention background, mention studies (if any) indicating the same.

2. Figure 4 : It will be more clear, If the symbol legends can be place on the figure itself.

3. Page 30 line 616: Please specify the group you are talking about.

4. Page 32: Conclusion: I have the same concern, how are you justifying the relationship between maternal behavior and hypoxia in pups? Or you are concluding on the basis of transcription changes, if so please mention. Also mention the same while introducing hypoxia pathway in the introduction by citing relevant studies.

Reviewer #3: The authors examined the effect of maternal licking and grooming (LG) during the first week of life on the timing of hearing onset in Wistar rat pups by different approaches. Among them auditory brainstem responses (ABR), tracking of eye opening (EO), micro-CT X-ray tomography, and gene expression analysis in 5 different brain regions. The authors found statistically significant increases in the relative mRNA levels of four genes involved in neurotrophin signaling in auditory brain regions from pups of different LG backgrounds. Nevertheless, the authors' hypothesis that LG backgrounds affect the timing of ABR onset, EO, and the relative mRNA levels of genes involved in the hypoxia-sensitive pathway has not been confirmed.

This paper is well presented and has good potential. The topic of the manuscript is interesting because the disruption of infant-mother interaction (as seen in low-LG mothers and some animal stress models) results in multiple delayed negative effects on behavioral phenotype and cognitive performance.

Of note, that the number of such studies at several time points is limited. There are, however, some points that need attention.

1. The number of animals used in some analysis is small, at least 6 - 8 (from at least 4 different litters) animals should have been used.

2. Different regions were used in the study: cochlear nucleus, pons (ventral brainstem containing the acoustic stria), inferior colliculus, temporal cortex (here referred to as auditory cortex), and occipital cortex (here referred as visual cortex). Please describe in more detail how they were dissected (indicating the required coordinates according to the atlas of the brain).

3. For PCR analysis, only one reference gene was used. Although this gene is often used as a housekeeping gene, have you checked that it is not differentially expressed in these brain regions?

7. PLOS authors have the option to publish the peer review history of their article (what does this mean?). If published, this will include your full peer review and any attached files.

Reviewer #1: No

Reviewer #3: No

---

## [Author Response · Author response to Decision Letter 1]

22 Jul 2020

Reviewer #1: The authors have addressed all the major concerns. And the manuscript is acceptable in its current form. However, I have a few minor comments as follows:

1. Page 3 line 62: Is there a proof of maternal LG behavior leading to hypoxia or oxygen deficiency in pups, please mention background, mention studies (if any) indicating the same.

Authors reply: Thank you for raising this point. We found two articles that are relevant to document a link between maternal LG behavior and oxygen deficiency in pups. We added the new text and cited the references on page 3 lines 63-66.

2. Figure 4 : It will be more clear, If the symbol legends can be place on the figure itself.

Authors reply: Thank you for the suggestion. We have added symbol legends in Figure 4.

3. Page 30 line 616: Please specify the group you are talking about.

Authors reply: Thank you for raising this issue. We have revised the text to clarify the LG groups on page 30 lines 623-628.

4. Page 32: Conclusion: I have the same concern, how are you justifying the relationship between maternal behavior and hypoxia in pups? Or you are concluding on the basis of transcription changes, if so please mention. Also mention the same while introducing hypoxia pathway in the introduction by citing relevant studies.

Authors reply: See our response to points 1 and 3 above. We have been careful in the discussion. For example, in page 30 lines 628 to 634 we state: ”Assuming that transcriptional down regulation of Egln1 causes reduced levels of mRNA transcripts and Phd2 in cells of the auditory system, this result would suggest that cells in the auditory system of neonate rats may have a differential sensitivity to oxygen challenges during postnatal development. This interpretation is consistent with results that showed higher vulnerability of P7 compared to P1 rats when exposed to hypoxia-ischemia, and with studies that tracked long-term maternal behavioral, learning and auditory deficits in rats exposed to perinatal anoxia at birth [16, 48, 49].” To address your concern in the conclusion, we state more carefully (page 32 lines 680 and 681: “The results of a gene expression screen suggest that sensitivity to hypoxic challenge might be widespread in the auditory system of neonate rats before hearing onset” We updated the abstract accordingly on page 2 lines 45 and 46.

Reviewer #3: The authors examined the effect of maternal licking and grooming (LG) during the first week of life on the timing of hearing onset in Wistar rat pups by different approaches. Among them auditory brainstem responses (ABR), tracking of eye opening (EO), micro-CT X-ray tomography, and gene expression analysis in 5 different brain regions. The authors found statistically significant increases in the relative mRNA levels of four genes involved in neurotrophin signaling in auditory brain regions from pups of different LG backgrounds. Nevertheless, the authors' hypothesis that LG backgrounds affect the timing of ABR onset, EO, and the relative mRNA levels of genes involved in the hypoxia-sensitive pathway has not been confirmed.

This paper is well presented and has good potential. The topic of the manuscript is interesting because the disruption of infant-mother interaction (as seen in low-LG mothers and some animal stress models) results in multiple delayed negative effects on behavioral phenotype and cognitive performance.

Of note, that the number of such studies at several time points is limited. There are, however, some points that need attention.

1. The number of animals used in some analysis is small, at least 6 - 8 (from at least 4 different litters) animals should have been used.

Authors reply: Thank you for your comment. In ideal conditions we would have liked to process more samples in the gene screen experiments. To adapt our study to the constraints of our budget we decided for triplicate samples from pups of one, two or three litters as described in the methods.

2. Different regions were used in the study: cochlear nucleus, pons (ventral brainstem containing the acoustic stria), inferior colliculus, temporal cortex (here referred to as auditory cortex), and occipital cortex (here referred as visual cortex). Please describe in more detail how they were dissected (indicating the required coordinates according to the atlas of the brain).

Authors reply: Thank you for your comment. We have updated the Gene Expression Methods section as requested (page 10 lines 230-237) with a brief description of the landmarks used to dissect brainstem structures according to Rodriguez-Contreras et al. (2014), and the approximate bregma/lambda coordinates according to the most up to date atlas of the neonate rat brain (Khazipov et al. 2015) in S1 Table.

3. For PCR analysis, only one reference gene was used. Although this gene is often used as a housekeeping gene, have you checked that it is not differentially expressed in these brain regions?

Authors reply: Thank you for raising this point. We were careful to check the housekeeping gene across all ages and LG conditions, serving as an internal control for the PCR analysis. It is important to note that we decided to compare mRNA levels for each gene as a fold change with respect to P0. For the sake of transparency we included the fold change data for the housekeeping gene Actb in Table 4, and revised the text on page 22 lines 498-501.

---

## [Decision Letter · Decision Letter 2]

6 Aug 2020

PONE-D-20-07005

Defining the relationship between maternal care behavior and sensory development in Wistar rats: auditory periphery development, eye opening and brain gene expression

PONE-D-20-07005R2

Dear Dr. Rodriguez-Contreras,

We’re pleased to inform you that your manuscript has been judged scientifically suitable for publication and will be formally accepted for publication once it meets all outstanding technical requirements.

Kind regards,

Giuseppe Biagini, MD

Academic Editor

PLOS ONE

Additional Editor Comments (optional):

Reviewers' comments:

Reviewer's Responses to Questions

**Comments to the Author**

1. If the authors have adequately addressed your comments raised in a previous round of review and you feel that this manuscript is now acceptable for publication, you may indicate that here to bypass the “Comments to the Author” section, enter your conflict of interest statement in the “Confidential to Editor” section, and submit your "Accept" recommendation.

Reviewer #1: All comments have been addressed

Reviewer #3: All comments have been addressed

2. Is the manuscript technically sound, and do the data support the conclusions?

Reviewer #1: Yes

Reviewer #3: Yes

3. Has the statistical analysis been performed appropriately and rigorously? 

Reviewer #1: Yes

Reviewer #3: Yes

4. Have the authors made all data underlying the findings in their manuscript fully available?

Reviewer #1: Yes

Reviewer #3: Yes

5. Is the manuscript presented in an intelligible fashion and written in standard English?

Reviewer #1: Yes

Reviewer #3: Yes

6. Review Comments to the Author

Reviewer #1: The authors have addressed all the points raised and I think the manuscript is acceptable in its current form.

Reviewer #3: All corrections have been addressed by the authors. The manuscript is suitable for publication in its current form.

7. PLOS authors have the option to publish the peer review history of their article (what does this mean?). If published, this will include your full peer review and any attached files.

Reviewer #1: No

Reviewer #3: No

---

## [Editor Report · Acceptance letter]

11 Aug 2020

PONE-D-20-07005R2 

PONE-D-20-07005
Defining the relationship between maternal care behavior and sensory development in Wistar rats: auditory periphery development, eye opening and brain gene expression 

Dear Dr. Rodriguez-Contreras:

I'm pleased to inform you that your manuscript has been deemed suitable for publication in PLOS ONE. Congratulations! Your manuscript is now with our production department. 

Kind regards, 

on behalf of

Dr. Giuseppe Biagini 

Academic Editor

PLOS ONE